# Encouraging pro-environmental behavior: Do testimonials by experts work?

**Olesya M. Savchenko**[1☉]*, **Leah H. Palm-Forster**[2☉], **Lusi Xie**[3], **Rubait Rahman**[4], **Kent D. Messer**[2]

**1** Food and Resource Economics Department, University of Florida, Gainesville, Florida, United States of America, **2** Department of Applied Economics and Statistics, University of Delaware, Newark, Delaware, United States of America, **3** Department of Agricultural and Applied Economics, University of Georgia, Athens, Georgia, United States of America, **4** Department of Agricultural, Food, and Resource Economics, Michigan State University, East Lansing, Michigan, United States of America

☉ These authors contributed equally to this work.
* olesya.savchenko@ufl.edu

## Abstract

Using non-pecuniary interventions to motivate pro-environmental behavior appeals to program administrators seeking cost-effective ways to increase adoption of environmental practices. However, all good-intended interventions should not be expected to be effective and reporting when interventions fail is as important as documenting their successes. We used a framed field experiment with 308 adults from the Mid-Atlantic in the United States to test the effectiveness of an expert testimonial in encouraging adoption of native plants in residential settings. Though studies have found testimonials to be effective in other contexts, we find that the video testimonial had no effect on residents' willingness to pay for native plants. Our analysis also shows that consumers who are younger, have higher incomes, and use other environmentally friendly practices on their lawns are more likely than other consumers to purchase native plants.

## Introduction

Policymakers and program managers are increasingly interested in identifying effective interventions to promote pro-environmental behavior [1, 2]. Growing evidence suggests that non-pecuniary behavioral interventions have the potential to motivate energy and water conservation [3–8], lead to more environmentally friendly food choices [9], improve recycling efforts [10], and reduce nonpoint source pollution [11]. However, gaps remain in our understanding of the effectiveness of specific interventions in motivating pro-environmental behavior [12, 13]. Based on a review of 160 experiment-based studies of environmental interventions, Byerly et al. [12] identified areas in need of future research to inform the design of policies and programs, including testing the effectiveness of interventions used in other contexts in promoting pro-environmental behavior. Behavioral interventions such as information provision [14, 15], peer comparisons [16], social norms [2, 17], and framing [18, 19] have been extensively studied. Less is known about testimonials such as opinions and recommendations provided by experts and influencers who speak positively about a pro-environmental product or practice.

**Data Availability Statement:** All data files are available from the OSF database (DOI 10.17605/OSF.IO/AWVQY). https://osf.io/awvqy.

**Funding:** Funding for this paper is from the National Science Foundation EPSCoR Grant No. 1757353 and the State of Delaware. (LPF, KDM, OMS) The funders had no role in study design, data collection and analysis, decision to publish, or preparation of the manuscript.

**Competing interests:** The authors have declared that no competing interests exist.

Several studies have found that testimonials are effective in some contexts [20–22]. In terms of cost-effectively promoting demand for consumer goods [23–28], they have been called the "workhorse selling tool" that never goes out of vogue [23, p 29]. They have also been identified as successful in promoting positive changes in health-related [20, 29, 30], child safety [31], and mental health [32] behaviors. Testimonials by experts who are seen as credible have been particularly effective in improving the believability of messages [33, 34], and several studies have found that audio-visual testimonials are more effective than testimonials communicated by text and photos [35, 36].

A few studies have explored the impacts of testimonial-based interventions on the adoption of pro-environmental behavior. Elgaaied-Gambier et al. [37] and He et al. [38] showed that having an endorser in printed advertisements had a positive impact on the intent of French consumers and students in China, respectively, to choose positive environmental behaviors. Studies that have examined the role of celebrity endorsements have produced mixed results. Olmedo et al. [39] and Ellis et al. [40] found that celebrity endorsements were largely ineffective in inducing pro-environmental behavior, while Ho et al. [41] demonstrated that a celebrity endorsement combined with an information campaign led to a 25% reduction in the use of plastic items among students in Vietnam. These studies provide valuable insights but present challenges when applying their findings to pro-environmental behavior in other countries, such as the United States. We extend the existing body of knowledge by conducting a revealed preference study of U.S. consumers to test the effect of an expert testimonial on willingness to pay (WTP) for native plants, which offer a suite of environmental benefits when planted in residential landscapes.

An important issue associated with experiment-based studies of behavioral interventions designed to increase pro-environmental behavior is the absence of statistical power analyses to determine adequate sample sizes and appropriate randomization [12]. The results of underpowered studies, consequently, can be misleading for policymakers and program managers when they fail to detect existing effects and overstate detected effects [42]. Therefore, randomized controlled experiment designs and sufficient sample sizes are needed to establish causal relationships between specific interventions and pro-environmental behavior [12, 43].

In this paper, we contribute to the existing literature related to testimonials–broadly defined as opinions and recommendations expressed by experts or influencers encouraging a behavior–by evaluating the effectiveness of an expert testimonial in promoting adoption of pro-environmental behavior using a randomized experiment design. We test the effect of viewing a video testimonial from an expert on participants' WTP for native plants in an incentive-compatible framed field experiment by recruiting 308 adults from the U.S. Mid-Atlantic region. We also present results from our power analyses so that the study findings and its implications for program design can be interpreted in an appropriate context.

Pro-environmental behaviors generally are defined as behaviors that reduce the actors' negative impacts on the environment [44, 45]. In this study, we focus on a specific residential pro-environmental behavior–the purchase of native plants. Native plants provide numerous environmental services, including supporting biodiversity [46], and they are an environmentally friendly best management practice for residential landscaping. They require less water and fewer pesticides and fertilizers than do conventional lawns [47], and reduce losses of biodiversity in urban and suburban landscapes by attracting a wide variety of insects and birds [46]. And, like all plants, they sequester carbon [48] and reduce runoff.

An important feature of our experiment is that it is non-hypothetical, presenting participants with opportunities to purchase native plants for their own use. This design allows us to examine the effect of an expert's video testimonial in an active market setting in which participants exchange real money for real goods, thus revealing their true preferences [49].

## Materials and methods

### Experiment design

We designed an incentive-compatible single-bounded dichotomous-choice framed field experiment to test the effect of a video testimonial about the benefits of native plants provided by an expert on WTP for Husker Red (*Penstemon digitalis*), a perennial plant native to the U.S. Mid-Atlantic region. The experiment, which took approximately 15 minutes to complete, was conducted in April and May 2018 at three locations in the U.S. Mid-Atlantic region–a large community event, a super-regional shopping mall frequented by more than 20 million shoppers each year, and an ice cream shop. The locations were selected because they provided access to a diverse pool of participants.

Signs and flyers were used to recruit potential participants 22 years of age or older who make management decisions about the landscape around their home in a two-part experiment. Prior to making those decisions, each participant was endowed with a $10 participation payment that was theirs to keep–they could also use this money to purchase plants. This study was approved by the University of Delaware Institutional Review Board (Project ID 1213674–3). Each participant was assigned an iPad Pro with headphones after reading and signing an informed consent statement to participate in the experiment. Participants completed the experiment asynchronously and were not permitted to communicate with other participants present at the same time. The experiment (control and treatment versions), the choice questions, and a short survey were presented electronically via the iPad screens.

To test the effect of the expert testimonial on their WTP for plants, participants were randomly assigned to either the non-video control group or the video-testimonial treatment group. Random assignment of participants to the control and testimonial treatments was implemented within the experiment platform that was developed using Willow, a Python framework for programming economic experiments [50]. Willow also did the random selection of which of the price treatments was implemented. Before making their WTP choices, participants received written information about the benefits of native plants (see Fig 1). Participants in the treatment group viewed the 90-second video testimonial in which the benefits of native plants were described by Dr. Doug Tallamy, a well-known professor of entomology and wildlife ecology at a local land-grant university, who is a highly regarded expert in the field and author of a best-selling book about conservation and landscape ecology. A transcript of the video testimonial and a web link to the video can be found in S1 Appendix of this paper. In the experiment, participants in the treatment group were required to watch the entire video before they could proceed with the experiment.

During the first portion of the experiment, participants were given the opportunity to purchase a bundle of three ready-to-plant landscape plugs of Husker Red. The iPads presented participants with a description of the plant (see Fig 2) and a picture of how it would look once fully grown. Participants were asked to indicate their willingness to purchase the bundle at a stated price in three dichotomous-choice (Yes/No) questions that presented different prices: $0, $3, and $6 per bundle. We included a free plant bundle ($0 price) decision because the second stage of the experiment, which is not reported here, was designed to analyze how self-reported plantings were affected by receiving the plants for free versus purchasing them. The prices were presented in random order to avoid potential ordering effects.

To ensure incentive-compatibility, participants were told that only one of the three purchase decisions would be binding, and that the binding decision would be randomly selected at the end of the study to determine participants' take-home cash and whether they purchased the native plants. This mechanism ensured that participants' dominant strategy was to answer 'Yes' whenever the posted price did not exceed their maximum WTP for the bundle. After

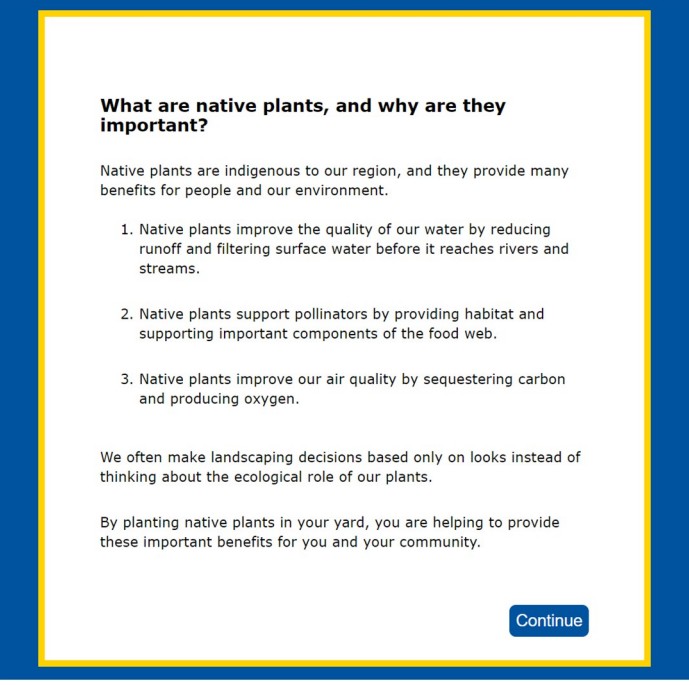

**Fig 1. Written information provided to participants about the benefits of native plants.**

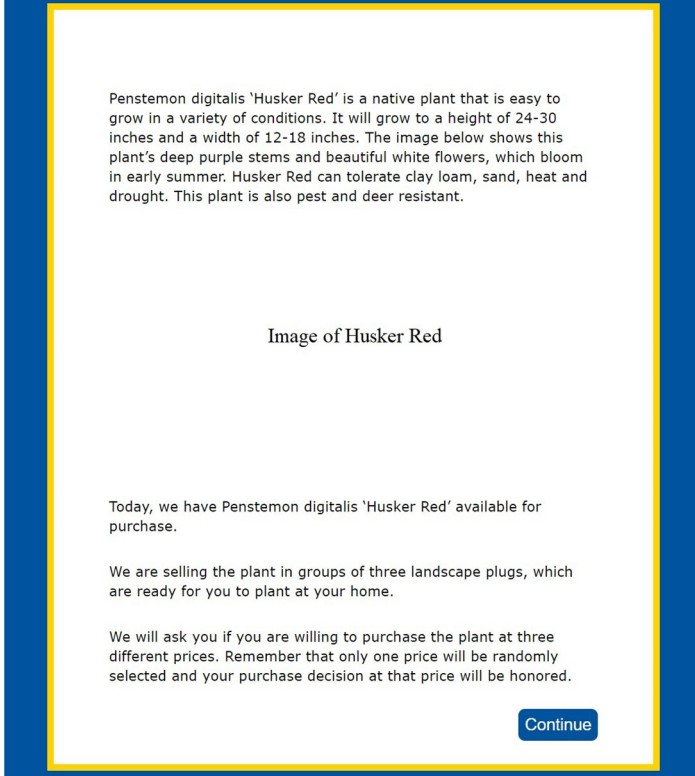

**Fig 2. Description of the native plant offered to experiment participants.**

answering the WTP questions, participants completed a short survey that collected information on their socio-demographic characteristics and their lawn care practices.

Once they completed the survey, the program randomly selected one of the prices (decisions) as binding and notified participants of their outcomes. Participants who indicated 'Yes' in the selected decision purchased the bundle of plants at that price, which was deducted from the $10 participation payment; they received the plant bundle and the balance of the participation payment. Participants who indicated 'No' did not purchase plants and received the entire $10 participation payment. As participants exited the experiment area, they returned the iPads and received their take-home funds and plants purchased, if any. A detailed experiment protocol and copy of the survey are provided in S2 Appendix.

Before conducting the experiment, we performed an ex-ante power analysis to determine the required sample size for the full experiment, which consisted of three treatment arms and two stages (in this paper, we present results of one treatment from the first stage of the experiment). The power analysis was conducted via simulation following the approach described by Feiveson [51] so that we could model the multi-stage structure of the experiment and account for some participants exiting the experiment after each stage. To implement the simulation, we used valuations of native plants collected via an experimental auction by members of our research team [18]. The final experimental design required 345 participants to reliably detect the effect of the video-testimonial on mean WTP for native plants between the treatment and control groups. To conduct the simulation, we assumed a standardized effect size of d = 0.30 with 80% power (i.e., $\beta = 0.80$ and $\alpha = 0.05$). Because we obtained data from only 308 participants, we also conducted an ex-post power analysis to understand the implications for our analysis. We present additional details about the ex-post power analysis in the results section.

## Data

In the experiment, we collected valid data from 308 participants. According to participants' self-reported zip codes, about 72.8% of the participants resided in Delaware, 9.1% of the participants resided in Pennsylvania, and 7.4% of the participants resided in Maryland. Table 1 provides summary statistics of the participants' socio-demographic characteristics. On average, participants in the experiment were 41 years old, 44% were male, and 42% had one or more children under 18 years old in their households. Approximately 66% of the participants were employed, and about one-third of the participant households fell into each annual income category: less than $50,000; $50,000 to $100,000; and greater than $100,000. In terms of education, 67% of the participants had undergraduate and/or graduate degrees. Most lived in urban areas (87%) and owned their homes (68%). About 84% of the participants indicated that they used environmentally friendly practices to maintain their lawns.

Two sample t-tests showed that the control and treatment groups were balanced in terms of most of the socio-demographic characteristics. However, compared to the control group, the treatment group included more males, participants with children, and annual incomes between $50,000 and $100,000. In addition, we conducted a test of joint orthogonality. The joint orthogonality test is complementary to t-tests on each individual variable. The purpose of a joint test of orthogonality is to check whether all socio-demographic variables are balanced (i.e., statistically indifferent) between the control and treatment groups. The results of the joint orthogonality test showed that the control and treatment groups are not balanced across all socio-demographic variables. We, therefore, control for socio-demographic characteristics in our models.

**Table 1. Summary statistics for socio-demographic variables by treatment.**

| Variable | Definition | Entire Sample | Control | Treatment | Difference (Control-Treatment) | Delaware (2018)[a] |
|---|---|---|---|---|---|---|
| Age | Participant's age in years in 2018 | **40.70** | **41.12** | **40.28** | **0.844** | **40.20** |
| | | (15.00) | (15.35) | (14.65) | (0.393) | (Median) |
| Male | Equals 1 for male participants | **0.438** | **0.494** | **0.383** | **0.110***** | **0.484** |
| | | (0.496) | (0.500) | (0.487) | (0.001) | |
| Children | Equals 1 when a child under 18 resided in the household | **0.416** | **0.370** | **0.461** | **-0.091***** | **0.252** |
| | | (0.493) | (0.483) | (0.499) | (0.005) | |
| Employed | Equals 1 when participant was employed | **0.659** | **0.662** | **0.656** | **0.006** | **0.586** |
| | | (0.474) | (0.473) | (0.476) | (0.835) | |
| Income$_{<\$50K}$ | Equals 1 when participant's income was less than $50,000 | **0.315** | **0.344** | **0.286** | **0.058*** | **0.383** |
| | | (0.465) | (0.476) | (0.452) | (0.056) | |
| Income$_{\$50K-\$100K}$ | Equals 1 when participant's income was between $50,000 and $100,000 | **0.354** | **0.299** | **0.409** | **-0.110***** | **0.322** |
| | | (0.478) | (0.458) | (0.492) | (0.000) | |
| Income$_{>\$100K}$ | Equals 1 when participant's income was greater than $100,000 | **0.331** | **0.357** | **0.305** | **0.052*** | **0.296** |
| | | (0.471) | (0.480) | (0.461) | (0.094) | |
| Education level | Equals 1 for participants who had a bachelor, graduate, or professional degree | **0.672** | **0.682** | **0.662** | **0.019** | **0.314** |
| | | (0.470) | (0.466) | (0.473) | (0.529) | |
| Urban | Equals 1 for participants living in urban areas | **0.873** | **0.890** | **0.857** | **0.032** | |
| | | (0.333) | (0.314) | (0.350) | (0.138) | |
| Homeowner | Equals 1 for participants who owned their homes | **0.679** | **0.662** | **0.695** | **-0.032** | |
| | | (0.467) | (0.473) | (0.461) | (0.291) | |
| Lawnsize$_{small}$ | Equals 1 when lawn size was less than 0.25 acres or participants did not have a lawn | **0.425** | **0.422** | **0.429** | **-0.006** | |
| | | (0.495) | (0.494) | (0.495) | (0.842) | |
| Lawnsize$_{medium}$ | Equals 1 when lawn size was between 0.25 and 1.0 acres | **0.448** | **0.435** | **0.461** | **-0.026** | |
| | | (0.498) | (0.496) | (0.499) | (0.428) | |
| Lawnsize$_{large}$ | Equals 1 when lawn size was greater than 1.0 acres | **0.127** | **0.143** | **0.110** | **0.032** | |
| | | (0.333) | (0.350) | (0.314) | (0.138) | |
| Lawn practice | Equals 1 when participants use environmentally friendly practices at home for the lawn | **0.844** | **0.831** | **0.857** | **-0.026** | |
| | | (0.365) | (0.375) | (0.350) | (0.277) | |
| N | | **308** | **154** | **154** | | **949,495** |

In the first three columns, mean values are bolded and standard deviations are in parentheses. The column labeled "Difference (Control-Treatment)" has mean values in bold and p values of t-tests in parentheses.

***, **, and * denote statistical significance at the 1%, 5%, and 10% level respectively.

[a]Source: 2018 American Community Survey: www.census.gov/acs/www/data/data-tables-and-tools/data-profiles/2018/.

A comparison of the sample to the general population of Delaware, where most of the participants resided (72.8%), showed that our sample was broadly representative of the state's population in terms of age, gender, and income. Our sample exceeded estimates for Delaware's population in terms of employment rate, number of children in the household, and education level.

## Analysis

The 308 participants in the experiment each made three 'Yes/No' purchase decisions, resulting in 924 observations. Given the binary nature of the decisions, we used a logistic regression to analyze the impact of the testimonial on purchases. Since any effect of the testimonial was

constant for the three decisions, we used a random effects logistic model with the following specification [52]:

$$P(D = 1) = \frac{\exp(\beta_0 + \beta_1 P + \beta_2 Testimonial + \boldsymbol{\gamma}' \mathbf{Z} + \varepsilon)}{1 + \exp(\beta_0 + \beta_1 P + \beta_2 Testimoninal + \boldsymbol{\gamma}' \mathbf{Z} + \varepsilon)} \tag{1}$$

where $D$ is a dummy variable indicating purchase decisions, $D = 1$ represents a 'Yes' decision, and $P$ is the posted price. *Testimonial* is a treatment dummy variable indicating whether a participant watched the video testimonial. Therefore, $\beta_2$ is the coefficient of interest that shows the treatment effect. $\mathbf{Z}$ is a vector of the socio-demographic variables presented in Table 1, and $\varepsilon$ is the individual stochastic error term. We estimated two models: a simple model that did not include the socio-demographic variables and a more complex one that included all of the socio-demographic variables to control for unbalanced variables between the control and treatment groups.

Using coefficient estimates from the logistic regression, we calculated the mean WTP for native plants following [53]:

$$WTP = -\frac{1}{\widehat{\beta}_1} \left( \widehat{\beta}_0 + \widehat{\beta}_2 Testimonial + \widehat{\boldsymbol{\gamma}}' \bar{\mathbf{Z}} \right) \tag{2}$$

in which *Testimonial* = 1 for WTP by treatment participants and *Testimonial* = 0 for WTP by control group participants. $\bar{\mathbf{Z}}$ includes the mean values (e.g., age, income) of the socio-demographic variables for the control and treatment groups. The models were estimated using maximum likelihood, and mean values and standard errors of WTP were calculated using the delta method in Stata.

## Results

We first examine participants' purchasing decisions by comparing differences in the percentage of participants who were willing to purchase native plants at each posted price (percentage of 'Yes' responses). As shown in Fig 3, the percentage of participants willing to purchase the native plants decreased as the posted price increased. Only about 80% were willing to take the free native plants (price of $0). The fact that some participants did not want the plants likely

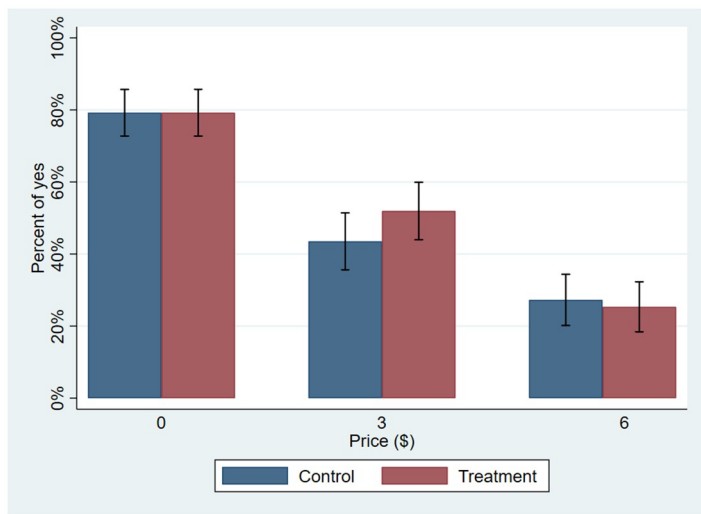

**Fig 3. Percentage of 'yes' responses at each price.**

reflects factors that limit adoption of native plants in general, including participants' landscaping preferences and their unwillingness to assume the maintenance and care requirements. These barriers and their implications for our study are explored further in the discussion section. About 43.5% and 27.3% of participants were willing to purchase the plants at $3 and $6, respectively.

As Fig 3 shows, the percentage of participants willing to take the free plants in the control and treatment groups is identical (79.2%). At a price of $3, more treatment participants (51.9%) than control participants (43.5%) were willing to purchase the native plants. At $6, slightly more control participants (27.3%) than treatment participants (25.3%) were willing to purchase the plants.

Table 2 reports the coefficients estimated from the random effects logistic regression used to identify factors that influenced participants' likelihood of purchasing native plants. We present the results for the simple model that estimated only the treatment effect (Model 1), the expanded model that included the full set of socio-demographic characteristics (Model 2), and the expanded model with an interaction term between price and testimonial (Model 3). Models 2 and 3 produced similar results.

The results of the regression indicate that price had a significantly negative effect on participants' likelihood to purchase the plants. This result is expected and is evident from Fig 3. The video testimonial had no effect on participants' decisions to purchase native plants as indicated by the positive but not significantly different from zero coefficient on the variable of interest, *Testimonial*. We also find that the interaction term between the *Price* and *Testimonial* was not statistically significant. In terms of socio-demographic characteristics, we find that older participants were less likely to purchase native plants than younger participants and that higher income ($50,000 or more per year) participants were more likely to purchase native plants than were lower income (less than $50,000 per year) participants. We further find that

**Table 2. Random effects logistic regression results.**

| | Model 1 | | Model 2 | | Model 3 | |
|---|---|---|---|---|---|---|
| | Coeff. | S.E. | Coeff. | S.E. | Coeff. | S.E. |
| Price | -0.654*** | (0.067) | -0.654*** | (0.067) | -0.648*** | (0.081) |
| Testimonial | 0.161 | (0.306) | 0.111 | (0.307) | 0.148 | (0.390) |
| Price*Testimonial | | | | | -0.012 | (0.093) |
| Age | | | -0.038*** | (0.012) | -0.038*** | (0.012) |
| Male | | | 0.160 | (0.307) | 0.160 | (0.306) |
| Children | | | -0.300 | (0.317) | -0.299 | (0.317) |
| Employed | | | -0.230 | (0.320) | -0.232 | (0.320) |
| Income$_{\$50K-\$100K}$ | | | 0.933** | (0.403) | 0.933** | (0.404) |
| Income$_{>\$100K}$ | | | 0.818* | (0.448) | 0.819* | (0.448) |
| Urban | | | -0.216 | (0.558) | -0.217 | (0.559) |
| Education level | | | 0.304 | (0.333) | 0.305 | (0.333) |
| Homeowner | | | 0.525 | (0.362) | 0.524 | (0.362) |
| Lawnsize$_{medium}$ | | | -0.099 | (0.342) | -0.098 | (0.342) |
| Lawnsize$_{large}$ | | | 0.790 | (0.506) | 0.788 | (0.506) |
| Lawn practice | | | 0.764* | (0.409) | 0.764* | (0.408) |
| Constant | 1.983*** | (0.301) | 2.070** | (0.808) | 2.051** | (0.805) |
| Log pseudolikelihood | | -504.192 | | -491.869 | | -491.858 |
| N | | 924 | | 924 | | 924 |

***, **, and * denote statistical significance at the 1%, 5%, and 10% level respectively.

participants who already used environmentally friendly practices to care for their lawns were more likely to purchase native plants. This result supports our notion that planting native species represents not only a way to express one's appreciation of the (local) natural environment but also a way to protect the environment (i.e., represents a pro-environmental behavior).

Using Eq 2 and the coefficient estimates from Model 2, we calculated the mean WTP for native plants for the control and treatment groups. Among the participants in the treatment group, the mean WTP for native plants was $3.28, 95% CI [2.66, 3.90], showing a $0.26 increase compared to the mean WTP of $3.02, 95% CI [2.39, 3.66] among the participants in the control group. This difference, however, is not statistically significant, which is consistent with the results of our earlier analysis. Collectively, our findings from the random effects logistic model and analysis of mean WTP values show that the behavioral intervention of providing participants with a video testimonial by an expert did not result in significant changes in participants' preferences for the pro-environmental behavior of buying native plants.

Failing to reject the null hypothesis of no treatment effect implies that either the experiment did not have enough statistical power to detect the treatment effect or there is no treatment effect [54]. We conducted an ex-post power analysis to determine the size of the testimonial effect that we could detect with 80% power (with alpha of 0.05) using our sample of 308 participants. The results of the power analysis vary depending on the underlying assumptions made about the distribution of WTP values. Our assumptions for the power analysis were guided by two previous studies of WTP for native plants. Using an experimental auction, Yue et al. [55] estimated the mean and standard deviation for plants labeled native/noninvasive as $2.79 and $2.90, respectively, with 5% of the bids at $0. Li et al. [18] estimated the mean and standard deviation to be $4.55 and $4.98, respectively, with 20% of the bids at $0. The results of these two studies suggest that the mean and standard deviation of values are similar in magnitude and that the distribution of values is skewed toward not paying anything at all. Considering the value distributions from these two studies, we conduct two ex-post power analyses. First, we calculate the minimum detectable effect size using Stata's 'power' command. Assuming a mean and standard deviation of $3.00, the minimum detectable effect size is $0.96, which equates to a standardized effect size of about 0.32. Since this calculation does not account for the structure of our data and the fact that we are not eliciting WTP values directly, we also use a simulation approach to calculate our power using the experimental auction bids elicited in Li et al. [18]. From this analysis, we find that our study is designed to detect an effect of $1.50 with 80% power.

Based on our power analyses, we cannot rule out the possibility that the testimonial has an effect below the $0.96 to $1.50 range. A larger sample size would allow us to detect smaller effects at the cost of recruiting and paying more participants. Researchers must weigh these costs against the potential benefits of detecting smaller effects. For example, our ex-post power analysis using the same simulation showed that we would have needed a sample of 6,000 participants to detect a $0.26 difference in mean WTP. Thus, it is possible that the video testimonial induced differences in mean WTP that were smaller than our study was able to detect, but such differences would not necessarily be economically meaningful for policy and program designs aimed at promoting pro-environmental behavior.

## Discussion

As policymakers, program managers, and practitioners seek effective ways of integrating behavioral interventions into policies and programs designed to promote pro-environmental behavior, a more robust evidence-base is needed that not only identifies what does work, but

importantly also identifies things that do not work or have limited impacts. This study examines the effectiveness of a testimonial (broadly defined as opinions and recommendations expressed by experts or influencers encouraging a behavior). Despite effective and widespread use of testimonials to market consumer goods and influence health-related decisions [20, 23–30, 32], little research has been conducted on using testimonials to encourage pro-environmental choices.

We use an incentive-compatible experiment involving actual consumer decisions to test the effect of expert testimonials on participants' decisions to purchase native plants for their residential lawns. This decision represents a pro-environmental landscaping choice. We compare the WTP for native plants between participants who viewed a video testimonial from a renowned expert in the field about the benefits of native plants and participants who did not view the video testimonial. Contrary to studies of testimonials used in other contexts [20, 29, 35], we find that the expert's video testimonial had no significant identifiable effect on WTP for native plants. We further find that consumers who are younger, have relatively higher incomes, and already use other environmentally friendly practices for their lawns are more likely than other consumers to purchase native plants. Our findings about the role socio-demographics and environmental preferences are consistent with previous studies exploring the determinants of pro-environmental behavior (see Blankenberg and Alhusen [56] for a review).

In this particular context, expert testimonials were not effective at motivating behavioral change, but this research design does not enable us to know *why* testimonials were not effective. Recent papers have emphasized the importance of identifying the key determinants limiting a desired behavior and then identifying the behavioral interventions that will likely be most effective at targeting those determinants [13] or the mechanisms of action through which changes in behavior occur [57]. One potential explanation as to why the testimonial failed is that it did not effectively target the key barriers limiting adoption of native plants. Previous studies have identified numerous barriers that limit planting of native plants in residential landscapes. These barriers include limited knowledge and information [58], lack of availability [59, 60], preferences for particular aesthetics [61], social/community norms [58, 62], direct costs and the time and effort required for maintenance and care [60], and characteristics of residents and their yards [63, 64]. Expert testimonials may be effective at addressing some of these barriers but not others.

Following the mapping of determinants and interventions by van Valkengoed et al. [13], we suggest that expert testimonials are generally designed to promote knowledge, change attitudes toward a pro-environmental behavior, and influence injunctive norms by having a respected expert communicating approval of an action. If knowledge, attitudes about integrating native plants in landscaping, and injunctive norms are the main barriers to adoption of native plants, our results indicate that testimonials are not effective at overcoming these barriers in this context. However, it is also possible that our testimonial intervention was not well-matched to the behavioral determinant limiting adoption of native plants.

Testimonials are not designed to overcome all behavioral barriers. For example, they are likely a poor match to address barriers related to the time and effort required to plant and maintain the native plants or to overcome a general dearth of efforts to protect the environment. In our experiment, we found that 20% of people did not want to take the native plant even when it is given away for free. One factor driving this outcome may be a perception of transaction costs associated with planting (e.g., finding or clearing the area where to plant, time and effort required to plant and care for the plants, etc.) among some participants. Transaction costs have been shown to be a barrier to adoption of pro-environmental behavior in other contexts, such as adoption of agricultural conservation practices [65, 66] and residential

landscape best management practices [67], among others. Further, without information about the level of the participants' commitment to environmental protection, it is unclear whether the transaction costs or lack of sufficient commitment to environmental protection to offset these costs [68, 69] may have led to the ineffectiveness of the testimonial treatment. These behavioral drivers are unlikely to be influenced by an expert testimonial treatment; therefore, our results may reflect a general mismatch between the behavioral intervention and the primary barriers limiting adoption of native plants.

Finally, while we used randomized assignment to the treatment and control groups, for the characteristics that were not balanced across these groups, other unidentified confounding factors (e.g., group differences in commitment to protect the environment) may have potentially precluded the identification of a statistically significant treatment effect. Further, the participants in our experiment tended to be more educated compared to the general population in the local area and could have already possessed some of the knowledge communicated in the testimonial, which could have led to the lack of a significant impact of the testimonial on participants' preferences.

Future studies can extend this research by identifying the primary barriers limiting adoption of native plants and testing behavioral interventions that are best suited to overcome these determinants [13, 57]. To further inform policymakers and program administrators, future studies can also test the efficacy of testimonial-based interventions in the context of other pro-environmental behavior in which the barriers are related to knowledge, attitudes, and injunctive norms–behavioral determinants that are more likely to be addressed by a testimonial intervention. Additionally, examining the effect of combining testimonials with other non-monetary behavioral interventions and/or financial incentives, or using larger samples to detect smaller effects of testimonials alone may be fruitful directions for future research.

Our results make several important contributions. First, we extend the literature examining non-monetary behavioral interventions on adoption of pro-environmental behavior by analyzing the effect of expert testimonials using a carefully designed, revealed preference experiment. Publishing null results and presenting results from power analyses are critical in supporting efforts to address publication bias that occurs when only statistically significant findings are reported [70]. Finally, our findings have practical implications, particularly for environmental organizations and government agencies considering using expert testimonials to increase engagement in their programs. Our results show that such efforts should consider whether the cost of creating and disseminating testimonials is worth the expected benefit, which could be small or non-existent in some contexts.

## Supporting information

**S1 Appendix. Transcript of the video testimonial provided by an expert.**
(DOCX)

**S2 Appendix. Experiment instructions and survey questions.**
(DOCX)

## Acknowledgments

We express sincere gratitude to all staff members at the Center for Experimental and Applied Economics at the University of Delaware for facilitating this research. We especially thank James Geisler for programming this experiment and Stephen Wemple for being a dedicated research assistant.

## Author Contributions

**Conceptualization:** Olesya M. Savchenko, Leah H. Palm-Forster, Lusi Xie, Rubait Rahman, Kent D. Messer.

**Data curation:** Olesya M. Savchenko, Leah H. Palm-Forster, Lusi Xie, Rubait Rahman.

**Formal analysis:** Olesya M. Savchenko, Leah H. Palm-Forster, Lusi Xie, Rubait Rahman.

**Funding acquisition:** Olesya M. Savchenko, Leah H. Palm-Forster, Kent D. Messer.

**Methodology:** Olesya M. Savchenko, Leah H. Palm-Forster, Lusi Xie, Rubait Rahman, Kent D. Messer.

**Writing – original draft:** Olesya M. Savchenko, Leah H. Palm-Forster, Rubait Rahman.

**Writing – review & editing:** Olesya M. Savchenko, Leah H. Palm-Forster, Lusi Xie, Kent D. Messer.

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
