## [Decision Letter · Decision Letter 0]

23 Nov 2022

PONE-D-22-27253Encouraging pro-environmental behavior: do testimonials by experts work?PLOS ONE

Dear Dr. Savchenko,

Thank you for submitting your manuscript to PLOS ONE. After careful consideration, we feel that it has merit but does not fully meet PLOS ONE’s publication criteria as it currently stands. Therefore, we invite you to submit a revised version of the manuscript that addresses the points raised during the review process. After careful reading of the paper and the review letters, I would like to give you the opportunity to revise your manuscript, particularly according to the comments of reviewer 1. However, I expect you respond in detail to reviewer 2's objections in a statement and to present their arguments against each individual point. For example, you could deal with certain points of criticism offensively and point them out in the limitations.

We look forward to receiving your revised manuscript.

Kind regards,

Dr. Florian Follert

Academic Editor

PLOS ONE

Journal Requirements:

2. Please provide additional details regarding ethical approval in the body of your manuscript. In the Methods section, please ensure that you have specified the name of the IRB/ethics committee that approved your study.

5. We note that Figure 2 and S2 in your submission contain copyrighted images. All PLOS content is published under the Creative Commons Attribution License (CC BY 4.0), which means that the manuscript, images, and Supporting Information files will be freely available online, and any third party is permitted to access, download, copy, distribute, and use these materials in any way, even commercially, with proper attribution. For more information, see our copyright guidelines: http://journals.plos.org/plosone/s/licenses-and-copyright.

a. You may seek permission from the original copyright holder of Figure 2 and S2 to publish the content specifically under the CC BY 4.0 license. 

6. We note that Figure S2 includes an image of a participant in the study. 

Reviewers' comments:

Reviewer's Responses to Questions

**Comments to the Author**

1. Is the manuscript technically sound, and do the data support the conclusions?

Reviewer #1: Yes

Reviewer #2: Yes

2. Has the statistical analysis been performed appropriately and rigorously? 

Reviewer #1: Yes

Reviewer #2: Yes

3. Have the authors made all data underlying the findings in their manuscript fully available?

Reviewer #1: Yes

Reviewer #2: Yes

4. Is the manuscript presented in an intelligible fashion and written in standard English?

Reviewer #1: Yes

Reviewer #2: Yes

5. Review Comments to the Author

Reviewer #1: REVIEW OF PONE-D-22-27253

"Encouraging pro-environmental behavior: Do testimonials by experts work"

In their research, the authors tested the effectiveness of an expert’s video testimonial in promoting people’s willingness to pay for native plants. In their field experiment with 308 volunteers, the authors did not find the expected effect of the testimonial. The authors nevertheless believe that, in addition to reports of successful behavior-change interventions, reports of negative results (i.e., indicating ineffective interventions) are also worth knowing about.

The paper is well-written and conventionally organized. The research objective is explicitly stated (see p. 4), and the experimental trial is properly conducted and reported. On the negative side, I find the focus in the Introduction on nudging misleading, as nudges are not only non-monetary behavioral interventions. The discussion is scant and should address some more alternative explanations for why the intervention could have failed. Moreover, some of the technicalities are not spelled out in enough detail. Although ineffective interventions leave much room for speculation on why they failed, this research nevertheless offers a valuable contribution to the literature on the promotion of pro-environmental behavior. In the following, I will elaborate on my appraisal in more detail.

Nudges are not only non-monetary influences on behavior. Rather, the term actually refers to aspects of the “choice architecture” that either facilitate the target behavior or impede the behavioral alternative (see, e.g., Taube & Vetter, 2019). Such facilitating/impeding structural factors are supposed to modify the boundary conditions in which a behavior or a decision takes place. They can and should be separated from providing information in all its forms (e.g., instructing, advertising, and messaging; see Osbaldiston & Schott, 2012). Whereas facilitating factors originate from rearranging the conditions in which a behavior takes place (e.g., by making something more accessible), the testimonial in this study provides sound reasons for some specific gardening practices (i.e., why information).

Another issue worth considering is the specific pro-environmental behavior under investigation: the procuring of a specific native plant. Are its “costs” a possible explanation for its relative lack of malleability (see, e.g., Kaiser et al., 2021). If the behavior is obstructed by costs (e.g., plant care effort and time demands) or the availability of an area that is ready for landscaping, only people with a comparatively strong commitment to environmental protection (i.e., with strong environmental attitudes) would respond favorably to the testimonial. Without knowledge about people’s attitudes and the relative costs of this particular behavior, we cannot rule out the challenging nature of gardening in general or the insufficient environmental attitude of the experimental group as alternative explanations of the intervention’s inefficacy (see Kaiser et al., 2010).

The differences between the treatment and control groups speak of not fully comparable subsamples and a somewhat failed random allocation (see Table 1). These different subsample characteristics could have biased the findings of this research. In their Model 2, the authors statistically controlled for such confounding influences (see Table 2). Nevertheless, there might still be other—unidentified—confounding factors (e.g., group differences in environmental attitude) that could have masked the efficacy of the testimonial.

Table 1 additionally speaks of some nonresponse error. People who volunteered for this research differed from people who did not volunteer, a difference that can be seen in the sample’s remarkable departure from the population in Delaware in educational level. The much higher education level of the sample could alternatively explain the inefficacy of the testimonial. Due to their advanced educational levels, participants might already possess the knowledge conveyed in the testimonial, which could account for the inefficacy of the testimonial.

In the following, I list some more issues that also need amendment before the paper will be ready for publication:

• Please specify the effect size measure (d, g, r) that you report (e.g., p. 8).

• What algorithm was used for the random presentation of prices (p. 7)?

• Also, specify the randomization procedure that was applied to randomly allocate participants to the experimental and control groups (p. 6). Doing so might help readers better understand the suboptimal allocation to groups (see Table 1).

• Provide the results of the t tests in more detail. Explain the “joint orthogonality tests,” their purpose, and your interpretation (p. 9).

• For completeness sake, in addition to the main effects of price and testimonial, I would also like to see the interaction between price and testimonial included in the random effects logistic regression model as an additional effect (p. 12).

• The text ends in a rather dull manner with the limitations of the research (p. 17). The paper should end with a major conclusion rather than with what Bem (1987) calls a “whimper.”

In conclusion, this research seems to be by and large competently conducted and reported. And despite the failed intervention, I recommend that this paper be published after it is revised, as it can make a valuable contribution to environmental protection research.

REFERENCES

Bem, D. J. (1987). Writing the empirical journal article. In M. P. Zanna & J. M. Darley (Eds.), The compleat academic: A practical guide for the beginning social scientist (pp. 171-201). McGraw-Hill.

Kaiser, F. G., Byrka, K., & Hartig, T. (2010). Reviving Campbell's paradigm for attitude research. Personality and Social Psychology Review, 14, 351-367.

Kaiser, F. G., Kibbe, A., & Hentschke, L. (2021). Offsetting behavioral costs with personal attitude: A slightly more complex view of the attitude-behavior relation. Personality and Individual Differences, 183, 111158.

Taube, O., & Vetter, M. (2019). How green defaults promote environmentally friendly decisions: Attitude‐conditional default acceptance but attitude‐unconditional effects on actual choices. Journal of Applied Social Psychology, 49, 721–732.

Osbaldiston, R., & Schott, J. P. (2012). Environmental sustainability and behavioral science: Meta-analysis of proenvironmental behavior experiments. Environment and Behavior, 44, 257–299.

Reviewer #2: The paper is well written, the analysis seems appropriately done and the results are clearly reported. Given the nature and execution of the experiment, the importance of the results and their contribution to the professional literature seem somewhat inflated. There are three basic problems which are explicated in the full review.

6. PLOS authors have the option to publish the peer review history of their article (what does this mean?). If published, this will include your full peer review and any attached files.

Reviewer #1: **Yes: **Florian G. Kaiser

Reviewer #2: No

---

## [Author Response · Author response to Decision Letter 0]

23 Mar 2023

Author's responses are attached in a word file named "Response to Reviewers." These responses are also presented below. 

Author’s response to comments by Reviewer #1: REVIEW OF PONE-D-22-27253

Title: "Encouraging pro-environmental behavior: Do testimonials by experts work"

Thank you for reading our manuscript and for the helpful comments and suggestions. By responding to your comments and the comments of the other referees, we believe our manuscript has significantly improved. You will find your original comments in bold, our response to your comments immediately afterwards, and then the sections of revised text, as appropriate. 

In their research, the authors tested the effectiveness of an expert’s video testimonial in promoting people’s willingness to pay for native plants. In their field experiment with 308 volunteers, the authors did not find the expected effect of the testimonial. The authors nevertheless believe that, in addition to reports of successful behavior-change interventions, reports of negative results (i.e., indicating ineffective interventions) are also worth knowing about.

The paper is well-written and conventionally organized. The research objective is explicitly stated (see p. 4), and the experimental trial is properly conducted and reported. On the negative side, I find the focus in the Introduction on nudging misleading, as nudges are not only non-monetary behavioral interventions. The discussion is scant and should address some more alternative explanations for why the intervention could have failed. Moreover, some of the technicalities are not spelled out in enough detail. Although ineffective interventions leave much room for speculation on why they failed, this research nevertheless offers a valuable contribution to the literature on the promotion of pro-environmental behavior. In the following, I will elaborate on my appraisal in more detail.

 Nudges are not only non-monetary influences on behavior. Rather, the term actually refers to aspects of the “choice architecture” that either facilitate the target behavior or impede the behavioral alternative (see, e.g., Taube & Vetter, 2019). Such facilitating/impeding structural factors are supposed to modify the boundary conditions in which a behavior or a decision takes place. They can and should be separated from providing information in all its forms (e.g., instructing, advertising, and messaging; see Osbaldiston & Schott, 2012). Whereas facilitating factors originate from rearranging the conditions in which a behavior takes place (e.g., by making something more accessible), the testimonial in this study provides sound reasons for some specific gardening practices (i.e., why information).

Thank you for this helpful comment. We edited the language in the abstract and in the manuscript to remove references to nudges and make it clear that we are focusing on non-monetary behavioral interventions and expert testimonials, specifically. 

Abstract: “Using non-pecuniary interventions to motivate pro-environmental behavior appeals to program administrators seeking cost-effective ways to increase adoption of environmental practices. However, we should not expect that all interventions are effective and reporting when interventions fail is as important as documenting their successes. We used a framed field experiment with 308 adults from the Mid-Atlantic in the United States to test the effectiveness of an expert testimonial in encouraging adoption of native plants in residential settings. Though studies have found testimonials to be effective in other contexts, we find that the video testimonial had no effect on residents’ willingness to pay for native plants. Our analysis also shows that consumers who are younger, have higher incomes, and use other environmentally friendly practices on their lawns are more likely than other consumers to purchase native plants.” 

[Page 2]: “Policymakers and program managers are increasingly interested in identifying effective interventions to promote pro-environmental behavior [1, 2]. Growing evidence suggests that non-pecuniary behavioral interventions have the potential to motivate energy and water conservation [3–8], lead to more environmentally friendly food choices [9], improve recycling efforts [10], and reduce nonpoint source pollution [11]. However, gaps remain in our understanding of the effectiveness of specific interventions in motivating pro-environmental behavior [12, 13]. Based on a review of 160 experiment-based studies of environmental interventions, Byerly et al. [12] identified areas in need of future research to inform the design of policies and programs, including testing the effectiveness of interventions used in other contexts in promoting pro-environmental behavior. Behavioral interventions such as information provision [14, 15], peer comparisons [16], social norms [17,18], and framing [19,20] have been extensively studied. Less is known about testimonials such as opinions and recommendations provided by experts and influencers who speak positively about a pro-environmental product or practice.”

Another issue worth considering is the specific pro-environmental behavior under investigation: the procuring of a specific native plant. Are its “costs” a possible explanation for its relative lack of malleability (see, e.g., Kaiser et al., 2021). If the behavior is obstructed by costs (e.g., plant care effort and time demands) or the availability of an area that is ready for landscaping, only people with a comparatively strong commitment to environmental protection (i.e., with strong environmental attitudes) would respond favorably to the testimonial. Without knowledge about people’s attitudes and the relative costs of this particular behavior, we cannot rule out the challenging nature of gardening in general or the insufficient environmental attitude of the experimental group as alternative explanations of the intervention’s inefficacy (see Kaiser et al., 2010).

Thank you for pointing this out. We included your suggestion as a potential explanation for the treatment’s ineffectiveness as follows: 

[Page 15]: “There may be several potential reasons that could explain why the expert testimonial treatment had no effect on participants’ WTP to purchase native plants. First, transaction costs associated with planting (e.g., finding or clearing the area where to plant, time and effort required to plant and care for the plants, etc.) could be perceived as high by some participants. Transaction costs have been shown to be a barrier to adoption of pro-environmental behavior in other contexts, such as adoption of agricultural conservation practices [57,58] and residential landscape best management practices [59], among others. Further, without information about the level of the participants’ commitment to environmental protection, it is unclear whether the transaction costs or insufficient strength of environmental attitudes to offset these costs [60,61] may have led to the ineffectiveness of the testimonial treatment.”

The differences between the treatment and control groups speak of not fully comparable subsamples and a somewhat failed random allocation (see Table 1). These different subsample characteristics could have biased the findings of this research. In their Model 2, the authors statistically controlled for such confounding influences (see Table 2). Nevertheless, there might still be other—unidentified—confounding factors (e.g., group differences in environmental attitude) that could have masked the efficacy of the testimonial.

We appreciate your concern and have included a discussion of this issue in the revised manuscript as follows: 

[Page 16]: “Second, while we used randomized assignment to treatments and controlled for the characteristics that were not balanced across the treatment and control groups in our analysis, other unidentified confounding factors (e.g., group differences in environmental attitudes) may have potentially precluded the identification of a statistically significant treatment effect.”

Table 1 additionally speaks of some nonresponse error. People who volunteered for this research differed from people who did not volunteer, a difference that can be seen in the sample’s remarkable departure from the population in Delaware in educational level. The much higher education level of the sample could alternatively explain the inefficacy of the testimonial. Due to their advanced educational levels, participants might already possess the knowledge conveyed in the testimonial, which could account for the inefficacy of the testimonial.

Good point. We included this potential driver of the inefficacy of the testimonial in the paper as follows: 

[Page 16]: “Finally, the participants in our experiment tended to be more educated compared to the general population in Delaware and could have already possessed some of the knowledge communicated in the testimonial, which could have led to the lack of a significant impact of the testimonial on participants’ preferences.”

In the following, I list some more issues that also need amendment before the paper will be ready for publication:

Please specify the effect size measure (d, g, r) that you report (e.g., p. 8).

We implemented our simulation to determine the sample size required to detect a 0.3 standard deviation difference in the mean WTP for native plants between our treatment (testimonial video) and control groups; i.e., we used a d family effect size measure of d=0.3. We have revised the text to read: 

[Page 8]: “The final experimental design required 345 participants to reliably detect the effect of the video-testimonial on mean WTP for native plants between the treatment and control groups. To conduct the simulation, we assumed a standardized effect size of d=0.30 with 80% power (i.e., β=0.80).”

 What algorithm was used for the random presentation of prices (p. 7)?

The experiment platform was developed using Willow, a Python framework for programming economic experiments (Weel and McCabe 2009). Random selection of the price treatments was implemented within the experiment platform. This is noted on page 6:

“Random assignment of participants to the control and testimonial treatments was implemented within the experiment platform that was developed using Willow, a Python framework for programming economic experiments [51]. Willow also did the random selection of which of the price treatments.”

51. Weel, J., McCabe, K., 2009. Willow: Experiments in Python. George Mason University. http://econwillow.sourceforge.net

 Also, specify the randomization procedure that was applied to randomly allocate participants to the experimental and control groups (p. 6). Doing so might help readers better understand the suboptimal allocation to groups (see Table 1).

When participants started the experiment, they were assigned to either the treatment or control group. Of the 308 participants, 154 were in each group. The experiment platform was developed using Willow, a Python framework for programming economic experiments (Weel and McCabe 2009). As noted above, the random assignment to treatments was implemented within the experiment platform.

 Provide the results of the t tests in more detail. Explain the “joint orthogonality tests,” their purpose, and your interpretation (p. 9).

Thank you for this suggestion. We have added a new column in Table 1. The new column shows the difference between control and treatment groups with p values of t-tests in parentheses. 

In addition, we conducted a test of joint orthogonality. The joint orthogonality test is complementary to t-tests on each individual variable. The purpose of a joint test of orthogonality is to check whether all socio-demographic variables are balanced (i.e., statistically indifferent) between the control and treatment groups. We first ran a logit regression where the dependent variable is the treatment dummy variable, and the independent variables are socio-demographic variables. With the regression results (say coefficient estimates are b1, b2, b3, …), we tested the joint hypothesis (b1=b2=b3=…b10=0). Since we rejected the null hypothesis (at a 1% significance level), we concluded that the control and treatment groups are not balanced across all socio-demographic variables. 

Please find the edits we made to the paper and the updated table 1 below. 

[Page 10]: “In addition, we conducted a test of joint orthogonality. The joint orthogonality test is complementary to t-tests on each individual variable. The purpose of a joint test of orthogonality is to check whether all socio-demographic variables are balanced (i.e., statistically indifferent) between the control and treatment groups. The results of the joint orthogonality test showed that the control and treatment groups are not balanced across all socio-demographic variables.”

[Page 9]: Table 1. Summary statistics for socio-demographic variables by treatment.

 For completeness sake, in addition to the main effects of price and testimonial, I would also like to see the interaction between price and testimonial included in the random effects logistic regression model as an additional effect (p. 12).

Thank you for this suggestion. We have added an interaction term between price and testimonial to the random effect logistic regression model, presented as Model 3 in Table 2. 

[Page 12]: “We present the results for the simple model that estimated only the treatment effect (Model 1), the expanded model that included the full set of socio-demographic characteristics (Model 2), and the expanded model with an interaction term between price and testimonial (Model 3). Models 2 and 3 produced similar results.”

[Page 13]: Table 2. Random effects logistic regression results.

 The text ends in a rather dull manner with the limitations of the research (p. 17). The paper should end with a major conclusion rather than with what Bem (1987) calls a “whimper.”

We appreciate this suggestion. We revised the manuscript so that it (hopefully) does not end with a “whimper”, but instead with a compelling summary of our findings. We provided suggestions for future research in the Results section: 

[Page 16]: “Future studies can extend this research by testing whether any of the factors described above could have an impact on the effectiveness of expert testimonials on pro-environmental behavior of buying plants. To further inform policymakers and program administrators, future studies can also test the efficacy of testimonial-based interventions in the context of other pro-environmental behavior, examine the effect of combining testimonials with other non-monetary behavioral interventions and/or financial incentives, or using larger samples to detect smaller effects of testimonials alone.” 

[Page 18]: “Our results make several important contributions. First, we extend the literature examining non-monetary interventions that rely on information provision on adoption of pro-environmental behavior by analyzing the effect of expert testimonials using a carefully designed, revealed preference experiment. Publishing null results and presenting results from power analyses are critical in supporting efforts to address publication bias that occurs when only statistically significant findings are reported [63]. Finally, our findings have practical implications, particularly for environmental organizations and government agencies considering using expert testimonials to increase engagement in their programs. Our results show that such efforts should consider whether the cost of creating and disseminating testimonials is worth the expected benefit, which could be small or non-existent based on the study’s results.”

 In conclusion, this research seems to be by and large competently conducted and reported. And despite the failed intervention, I recommend that this paper be published after it is revised, as it can make a valuable contribution to environmental protection research.

Thank you for this comment. We appreciate your valuable suggestions and believe that they have significantly improved our manuscript. We agree with you that publishing interventions that fail is just as important as documenting those that are successful. 

REFERENCES

Bem, D. J. (1987). Writing the empirical journal article. In M. P. Zanna & J. M. Darley (Eds.), The compleat academic: A practical guide for the beginning social scientist (pp. 171-201). McGraw-Hill.

Kaiser, F. G., Byrka, K., & Hartig, T. (2010). Reviving Campbell's paradigm for attitude research. Personality and Social Psychology Review, 14, 351-367.

Kaiser, F. G., Kibbe, A., & Hentschke, L. (2021). Offsetting behavioral costs with personal attitude: A slightly more complex view of the attitude-behavior relation. Personality and Individual Differences, 183, 111158.

Taube, O., & Vetter, M. (2019). How green defaults promote environmentally friendly decisions: Attitude‐conditional default acceptance but attitude‐unconditional effects on actual choices. Journal of Applied Social Psychology, 49, 721–732.

Osbaldiston, R., & Schott, J. P. (2012). Environmental sustainability and behavioral science: Meta-analysis of proenvironmental behavior experiments. Environment and Behavior, 44, 257–299.

 

Author’s response to comments by Reviewer #2: REVIEW OF PONE-D-22-27253

Title: "Encouraging pro-environmental behavior: Do testimonials by experts work"

Thank you for reading our manuscript and for your comments. You will find your original comments in bold, our response to your comments immediately afterwards, and then the sections of revised text, as appropriate. 

Reviewer’s summary

A field experiment is reported in which 308 participants, visiting either a community event or a shopping mall or an ice cream shop, were asked to indicate their willingness to pay $ 0, 3 or 6 for an attractive native plant for their garden, which price – if they would purchase – was deducted from a sure $ 10 reward for their participation. Both a control group and a treatment group of each about 150 subjects first received written information about the nature of the experiment and what they were expected to do. In their instruction the planting of native plants was described as promoting biodiversity, preventing water runoff and sequestering carbon dioxide. 

The treatment group was additionally presented with a 90-seconds’ video testimonial by a renowned plant-ecologist who essentially presented the same message in person. Main results indicate that there was no significant effect of the (additional) video presentation on subjects’ willingness to pay (at least about one dollar more than the control group) for the native plant offered. Besides there we some significant effects (e.g. of plant price, age, education) that could generally have been expected. The authors emphasize the importance of publishing negative effects of experimental treatments in the context of effectively encouraging pro-environmental behavior. And they conclude (a. o.) that “Our results make several important contributions.”

General evaluation

The paper is well written, the analysis seems appropriately done and the results are clearly reported. Given the nature and execution of the experiment, the importance of the results and their contribution to the professional literature seem somewhat inflated. 

 A fundamental problem in the design of the experiment is that all subjects were offered a $10 reward at the outset, and that their instruction implied that they would end up with the full $ 10 if they had expressed a zero willingness-to-pay for the bundle of three native plants or when they had expressed some WTP and were randomly assigned to the decision to purchase the plant for $ 0 (rather than $ 3 or 6). This may well have induced a bias for subjects to reduce any willingness to pay more than $ 0 for the plants.

We appreciate your concerns, however, we restfully disagree that this is a fundamental problem. In fact, providing compensation to subjects for participating in economic experiments is a widely accepted practice and a critical component of conducting experimental research that allows researchers to model behavior in non-hypothetical settings (Voslinsky and Azar, 2021, Davis et al. 2010). Research also suggests that providing compensation increases participation (Krawczyk, 2011, Abeler and Nosenzo, 2015; Gajic et al, 2012), especially in contexts when getting subjects to participate may be challenging (Kerr et al., 2012) and this can help reduce bias that may arise from having only voluntary participants.

In sum, we followed well-established practices in experimental economics and previous literature to design our experiment. Subjects in our experiment were endowed with $10 at the beginning of the experiment that they could spend on purchasing plants, if they wanted. While more participants were willing to get the plants for free, a substantial number of participants were willing to pay $3 or $6 (see Fig. 3). Following common protocols in experimental economics, participants were told that only one of the three purchase decisions would be binding, and that the binding decision would be randomly selected at the end of the study to determine participants’ take-home cash and whether they purchased the native plants. This mechanism ensured that participants had no incentive to misrepresent their WTP as their dominant strategy was to answer ‘Yes’ whenever the posted price did not exceed their maximum WTP for the bundle. In addition, since participants in both control and treatment groups received the same $10 as compensation, this compensation unlikely affected our ability to examine the impacts of testimonial video on participants’ willingness to pay, which was a key question of this research. 

References

Abeler J, Nosenzo D. Self-selection into laboratory experiments: pro-social motives versus monetary incentives. Experimental Economics. 2015 Jun;18:195-214.

Canavari M, Drichoutis AC, Lusk JL, Nayga Jr RM. How to run an experimental auction: A review of recent advances. European Review of Agricultural Economics. 2019 Dec 1;46(5):862-922.

Davis LR, Joyce BP, Roelofs MR. My money or yours: house money payment effects. Experimental Economics. 2010 Jun;13:189-205.

Gajic A, Cameron D, Hurley J. The cost-effectiveness of cash versus lottery incentives for a web-based, stated-preference community survey. The European Journal of Health Economics. 2012 Dec;13:789-99.

Kerr J, Vardhan M, Jindal R. Prosocial behavior and incentives: evidence from field experiments in rural Mexico and Tanzania. Ecological Economics. 2012 Jan 15;73:220-7.

Krawczyk M. What brings your subjects to the lab? A field experiment. Experimental Economics. 2011 Nov;14:482-9.

Voslinsky A, Azar OH. Incentives in experimental economics. Journal of Behavioral and Experimental Economics. 2021 Aug 1;93:101706.

 Another problem is that the encouraging message about buying native plants was essentially the same for the control group and the treatment group, the latter differing only in a repeat and some extension of the message by the plant ecologist in person on video. So, was there really a substantive difference in treatment between the two groups?

The goal of the experimental design was to determine whether a video expert testimonial had an impact on the pro-environmental behavior of purchasing native plants. To isolate this effect, participants in the control and treatment groups received the same information with the exception of the testimonial that was provided only to the treatment group. This set up provided for a clean identification of the impact of the testimonial because we varied only one feature of the information set provided to the two groups. Subjects were assigned to treatment and control groups randomly, which allowed us to estimate the causal effect of the testimonial treatment we were interested in isolating (Rubin, 1974). 

Reference

Rubin DB. Estimating causal effects of treatments in randomized and nonrandomized studies. Journal of educational Psychology. 1974 Oct;66(5):688.

 A third problem may have resided in the unexpected invitation to participate in an experiment, while subjects had arrived with the intention (whatever) to attend the community event, to go shopping or to get an ice cream, respectively. It would seem that under such circumstances subjects being offered $ 10 for their (brief?) participation were naturally inclined to read instructions quickly and go along the presented questions soon enough to obtain their $ 10 reward, either or not lowered by the price of the bundle of plants they had purchased (or rather: had to purchase) following the study design, whereafter they would continue their journey in accordance with their original plan of visit.

Subject recruitment for field experiments often takes place at venues that provide researchers with access to a diverse subject pool and real market environments (Canavari et al., 2019), such as the ones used to conduct our experiment. Participation in the experiment was voluntary and participants were informed of the time commitment associated with their participation. The experiment was administered by experienced researchers and trained administrators who sought to ensure that participants did not speed through the experiment and took the expected time in carrying out the tasks of the experiment. 

References

Canavari M, Drichoutis AC, Lusk JL, Nayga Jr RM. How to run an experimental auction: A review of recent advances. European Review of Agricultural Economics. 2019 Dec 1;46(5):862-922.

Some specific points

 The title of the paper is rather general. A more specific and informative version would be: ”Encouraging consumers to buy native plants: Do expert testimonials about ecological benefits work?”

Thank you for this suggestion. We prefer to keep a broader title for this paper because adoption of native plants in residential settings is a type of pro-environmental behavior. We have taken your comment into further consideration as we revised the abstract and introduction of the paper. In the revised manuscript, we are now more specific about the type of pro-environmental behaviors that are the focus of the research. Further, we believe that keeping the title as it currently stands will facilitate the ability of readers broadly looking for research related to pro-environmental behavior to find this paper via a general search as the implications are for more than just native plants. 

 The first sentence of the Abstract (and the first paragraph of the Introduction..) needs qualification. In many situations much depends on the various costs (whatever) for a person or household to adopt pro-environmental behavior, and upon other encouragement strategies such as product availability, relevant knowledge and abilities, and the like. [The authors cite Abrahamse’s (2019) book, rich in empirical information about ‘what works, what doesn’t and why’.] The last sentence of the abstract provides non-surprising information which is unrelated to the difference between experimental conditions.

Thank you for the suggestion. We have modified the first sentence of the abstract to clarify our focus on non-pecuniary interventions (expert testimonial, specifically) that may be effective at motivating pro-environmental behavior. While there are different factors that may drive the adoption of pro-environmental behavior, we felt that providing information about them in the introduction may be confusing to the readers as testing those factors is not the goal of this research. We edited the language of the abstract and the introduction to further clarify that this paper is focusing on non-monetary behavioral interventions. 

Although the results stated in the last sentence of the abstract may be intuitive, we prefer to include them as they are part of the findings of this research, in addition to the primary results that the testimonial did not have a significant impact on participants’ WTP for native plants. 

Abstract: “Using non-pecuniary interventions to motivate pro-environmental behavior appeals to program administrators seeking cost-effective ways to increase adoption of environmental practices. However, we should not expect that all interventions are effective and reporting when interventions fail is as important as documenting their successes. We used a framed field experiment with 308 adults from the Mid-Atlantic in the United States to test the effectiveness of an expert testimonial in encouraging adoption of native plants in residential settings. Though studies have found testimonials to be effective in other contexts, we find that the video testimonial had no effect on residents’ willingness to pay for native plants. Our analysis also shows that consumers who are younger, have higher incomes, and use other environmentally friendly practices on their lawns are more likely than other consumers to purchase native plants.” 

[Page 2]: “Policymakers and program managers are increasingly interested in identifying effective interventions to promote pro-environmental behavior [1, 2]. Growing evidence suggests that non-pecuniary behavioral interventions have the potential to motivate energy and water conservation [3–8], lead to more environmentally friendly food choices [9], improve recycling efforts [10], and reduce nonpoint source pollution [11]. However, gaps remain in our understanding of the effectiveness of specific interventions in motivating pro-environmental behavior [12, 13]. Based on a review of 160 experiment-based studies of environmental interventions, Byerly et al. [12] identified areas in need of future research to inform the design of policies and programs, including testing the effectiveness of interventions used in other contexts in promoting pro-environmental behavior. Behavioral interventions such as information provision [14, 15], peer comparisons [16], social norms [17,18], and framing [19,20] have been extensively studied. Less is known about testimonials, such as opinions and recommendations, provided by experts and influencers who speak positively about a pro-environmental product or practice.”

 On p. 5, line 109, a phrase like “.. an incentive-compatible single-bounded dichotomous-choice framed field experiment” sounds unnecessarily complicated. Readers might want to know what ‘incentive-compatible’ and ‘single-bounded’ practically mean. And ‘binary’ seems simpler than ‘dichotomous’.

We appreciate your concern. Much of this terminology is in-line with language common in experimental economics and is necessary to precisely describe the features of the experimental design. To help the reader, in the revised manuscript, we describe what incentive compatibility means as follows: 

[Page 7]: “To ensure incentive-compatibility, participants were told that only one of the three purchase decisions would be binding, and that the binding decision would be randomly selected at the end of the study to determine participants’ take-home cash and whether they purchased the native plants. This mechanism ensured that participants’ dominant strategy was to answer ‘Yes’ whenever the posted price did not exceed their maximum WTP for the bundle.”

 Note that data collection for the reported experiment was in 2018, i.e. at least four years ago. Why such delay?

We appreciate the concern and we would have preferred to have published this paper immediately after the data was collected. However, there were elements that delayed the final write-up of the paper, including the COVID pandemic. Ultimately, the time lag between data collection and publication had no impact on the quality of the research or this publication. 

 On p. 6, lines 122-123, sentence “Participants completed the experiment asynchronously and were not permitted to communicate with other participants present at the same time” raises questions about the practical setting in which subjects had to perform the requested task. See also general problem 3 above.

The experiment was administered by a team of trained administrators who ensured that all participants were following the same protocols. At each venue where the experiment took place, the research team set up tables with chairs for the participants and provided a display of the plants participants would be able to purchase. Each participant who voluntarily agreed to participate in the experiment was seated at an individual table and was provided an iPad to complete the experiment. The team of researchers and administrators who were running the experiment ensured that participants completed the experiment on their own, did not rush through the experiment, and answered any questions that participants had. Ensuring that each participant completed the experiment on their own was an important part of the experimental design so that we could collect individual preferences unaffected by the opinions of others. While allowing communication between participants would have been an option, we felt that our research design was cleaner by not permitting communication as we did not want spillover effects, such as when a person in the testimonial treatment talks about that information to a person in the control treatment. 

 Also on p. 6, lines 141-142: the three-point WTP-scale of $ 0, 3 and 6 may not have been sensitive or extensive enough to assess the possible added effects of the video-message following the written instruction.

This is an interesting comment. Future research should explore the sensitivity of WTP to price scale. 

 A nice aspect of the paper is the authors’ exercise about the (in-) sufficiency of statistical power to detect possible treatment effects. On p. 15, lines 284-285, the authors conclude that “Based on our power analyses, we cannot rule out the possibility that the testimonial has [or rather: ‘had’?] an effect below the $0.96 to $1.50 range.” If so, this might have meant that subjects were willing to pay (approximately) fifty cents more in response to the video-message. Would the authors have found this an interesting result worth publishing in a professional journal? [And note again general problem 2 above.]

We are being transparent about the size of effect this experiment was powered to detect. Given the statistical power of our experiment, we do not find that an expert testimonial has a significant impact on participants’ WTP. This finding is worth publishing as more robust literature is needed that not only identifies what types of behavioral interventions work, but importantly also identifies things that do not work or have limited impacts. Future research can explore smaller effect sizes with great sample sizes. However, researchers must weigh the costs of larger sample sizes against the potential benefits of detecting smaller effects. If our findings indicated that subjects were willing to pay fifty cents more in response to the video-message, we would have found this result worth publishing. 

 Experienced researchers in ‘encouraging pro-environmental behavior‘ might object to the strong generalization speaking from the passage on p. 16, lines 307-310: “Contrary to studies of testimonials used in other contexts […], we find that the expert’s video testimonial had no significant identifiable effect on WTP for native plants. Consequently, in pro-environmental contexts, testimonial-based interventions alone are not necessarily sufficient to motivate changes in behavior.” The generality of this conclusion does not seem entirely warranted.

The goal of this statement is to summarize our findings that are in contrast to the results found in other contexts. We employ a strong experimental design and are transparent about the size of the effects that can be detected in our experiment. We edited the last sentence of this passage to make this statement softer:

[Page 17]: “Consequently, in pro-environmental contexts, testimonial-based interventions alone may not necessarily be sufficient to motivate changes in behavior.”

It is not clear what is implied by the statement “experienced researchers in encouraging pro-environmental behavior.” The team of researchers that conducted this study has significant experience in researching a wide range of topics related to environmental behavior and environmental policies and has substantial experience in designing economic experiments.

---

## [Decision Letter · Decision Letter 1]

10 May 2023

PONE-D-22-27253R1

Encouraging pro-environmental behavior: do testimonials by experts work?

PLOS ONE

Dear Dr. Savchenko,

Thank you for submitting your manuscript to PLOS ONE. After careful consideration, we feel that it has merit but does not fully meet PLOS ONE’s publication criteria as it currently stands. Therefore, we invite you to submit a revised version of the manuscript that addresses the points raised during the review process.

Please be aware that we invited a new reviewer 3. I would like to invite you considering the comments by reviewer 1 and reviewer 3.

We look forward to receiving your revised manuscript.

Kind regards,

Florian Follert

Academic Editor

PLOS ONE

Journal Requirements:

Reviewers' comments:

Reviewer's Responses to Questions

**Comments to the Author**

1. If the authors have adequately addressed your comments raised in a previous round of review and you feel that this manuscript is now acceptable for publication, you may indicate that here to bypass the “Comments to the Author” section, enter your conflict of interest statement in the “Confidential to Editor” section, and submit your "Accept" recommendation.

Reviewer #1: All comments have been addressed

Reviewer #3: All comments have been addressed

2. Is the manuscript technically sound, and do the data support the conclusions?

Reviewer #1: Yes

Reviewer #3: Yes

3. Has the statistical analysis been performed appropriately and rigorously? 

Reviewer #1: Yes

Reviewer #3: Yes

4. Have the authors made all data underlying the findings in their manuscript fully available?

Reviewer #1: Yes

Reviewer #3: Yes

5. Is the manuscript presented in an intelligible fashion and written in standard English?

Reviewer #1: Yes

Reviewer #3: Yes

6. Review Comments to the Author

Reviewer #1: REVIEW OF PONE-D-22-27253.R1

"Encouraging pro-environmental behavior: Do testimonials by experts work?"

The authors did a good job revising their research report. I feel that most of my concerns from the last round of reviews were addressed in this revision. Still, there are some minor issues that I believe should be revisited once more. The places to be revisited are the following:

• On page 8, lines 176-177, and on page 15, line 281: Next to N, effect size, and power, the alpha-error should be explicitly stated here as well. This is necessary because the authors apply multiple alpha-error criteria throughout their text (see, e.g., the table notes of Tables 1 and 2).

• On page 14: Why are the expressions “Testimonial” and “Price and Testimonial” capitalized in line 259?

• In response to Reviewer #2’s criticism that the title was too general (“Encouraging pro-environmental behavior”), I suggest the authors seize the opportunity on page 14, lines 265-266, to strengthen their claim. Instead of “This result is intuitive since planting native species represents pro-environmental behavior,” I suggest they write something along the lines of “This result supports our notion that planting native species represents not only a way to express people’s appreciation of the (local) natural environment but also a way to protect the environment (i.e., a pro-environmental behavior).” More details on the argument can be found in Kaiser et al. (2013).

• On page 15, line 289: “skewed to the left” is less informative than “skewed toward not paying anything at all.”

• I suggest moving the text from page 14 (line 278) to page 17 (line 329) in the Discussion section: before line 353 (now on page 18).

• The DOI addresses are missing from several of the articles listed in the Reference list.

Conclusion. After revisiting these issues and amending the text along the suggested lines, I believe the research report will make a fine contribution to the literature. I hope the article will receive the attention it deserves.

REFERENCES

Kaiser, F. G., Hartig, T., Brügger, A., & Duvier, C. (2013). Environmental protection and nature as distinct attitudinal objects: An application of the Campbell paradigm. Environment and Behavior, 45, 369-398.

Reviewer #3: In this article, the authors conduct an experiment to examine whether an expert testimonial can motivate people to purchase native plants, a behaviour that is more pro-environmental than purchasing non-native plants. The authors found that there was no statistically significant difference in willingness to pay between the people who listened to the expert testimonial and the people in the control condition. The authors conclude that testimony-based interventions alone may not necessarily be sufficient to motivate changes in behaviour.

This paper reports an interesting field experiment with ecological validity. I think the focus on testimonials adds something new to the literature, and I appreciate the measure of behaviour and the relatively realistic research setting (compared to the traditional lab experiment). I am looking at this work for the first time now, but I realise the authors have already addressed many comments from the previous round of reviews, and seem to have done so thoroughly.

My addition to the discussion about this paper does not pertain to the methodology or the results, but more to how this research fits in with overall discussion on behaviour change interventions for pro-environmental behaviour. In recent years, we have seen many meta-analyses and review articles that tried to find out which interventions to promote pro-environmental behaviour are effective and which are not (e.g., Byerly et al., 2018; Mertens et al., 2022; Nisa et al., 2019). This approach of examining which interventions are effective overall has been criticized, since it is unlikely that particular interventions will always be effective whereas others will always be ineffective. Rather, the effectiveness of an intervention depends on the extent to which the intervention matches the target behaviour and the context in which the behaviour is implemented (van Valkengoed et al., 2022). This is broadly recognized within the field of behaviour change in general as well (e.g., Michie et al., 2018).

In general, I find that this research is not well embedded in this ongoing discussion in the literature, and does not really engage with any of the previously mentioned articles. In addition, this research seems to fall in line with the philosophy of trying to determine whether an intervention is effective or not, without considering the context in which the intervention is implemented, and whether there is a match between the intervention and the target behaviour. For example, the authors currently draw the conclusion that ‘in pro-environmental contexts, testimonial-based interventions alone may not necessarily be sufficient to motivate changes in behavior’. I would like to ask the authors to move away from these general conclusions about whether interventions do or do not work in general. Rather, I would like the authors to reflect more on why a testimonial-based intervention did not work in this specific context.

Following the procedure outlined by van Valkengoed et al., (2022), this would involve a more extensive discussion on what the possible drivers and barriers are to purchasing non-native plants, and an analysis of which of these barriers are and are not addressed by the intervention in question. Throughout the article, the authors already highlight several relevant barriers to purchasing native plants, such as landscaping preferences and being unwilling to assume the cost and/or care of the plant. Given the apparent presence of these barriers, it is not surprising to me that the testimonial intervention did not work in this context, since an expert testimonial does not address these specific barriers directly. There seems to be a mismatch between the intervention and the barriers to the target behaviour. I think this would be a fairer interpretation of the findings in this paper compared to drawing conclusions about testimonial-based interventions in general, especially since the authors themselves cite many papers in the introduction of this paper that highlight the effectiveness of testimony-interventions. As the paper currently stands, this leaves the reader wondering why the authors conclude that testimonials do not work for pro-environmental behaviours.

References

Byerly, H., Balmford, A., Ferraro, P. J., Hammond Wagner, C., Palchak, E., Polasky, S., . . . Fisher, B. (2018). Nudging pro-environmental behavior: evidence and opportunities. Frontiers in Ecology and the Environment, 16(3), 159-168. https://doi.org/10.1002/fee.1777

Mertens, S., Herberz, M., Hahnel, U. J. J., & Brosch, T. (2022). The effectiveness of nudging: A meta-analysis of choice architecture interventions across behavioral domains. Proc Natl Acad Sci U S A, 119(1). https://doi.org/10.1073/pnas.2107346118

Michie, S., Carey, R. N., Johnston, M., Rothman, A. J., de Bruin, M., Kelly, M. P., & Connell, L. E. (2018). From Theory-Inspired to Theory-Based Interventions: A Protocol for Developing and Testing a Methodology for Linking Behaviour Change Techniques to Theoretical Mechanisms of Action. Annals of Behavioral Medicine, 52(6), 501-512. https://doi.org/10.1007/s12160-016-9816-6

Nisa, C. F., Belanger, J. J., Schumpe, B. M., & Faller, D. G. (2019). Meta-analysis of randomised controlled trials testing behavioural interventions to promote household action on climate change. Nat Commun, 10(1), 4545. https://doi.org/10.1038/s41467-019-12457-2

van Valkengoed, A. M., Abrahamse, W., & Steg, L. (2022). To select effective interventions for pro-environmental behaviour change, we need to consider determinants of behaviour. Nat Hum Behav. https://doi.org/10.1038/s41562-022-01473-w

7. PLOS authors have the option to publish the peer review history of their article (what does this mean?). If published, this will include your full peer review and any attached files.

Reviewer #1: **Yes: **Florian G. Kaiser

Reviewer #3: No

---

## [Author Response · Author response to Decision Letter 1]

24 Jun 2023

Author’s response to comments by Reviewer #1: PONE-D-22-27253R1

Title: "Encouraging pro-environmental behavior: Do testimonials by experts work"

Thank you for reading our manuscript and for the helpful comments and suggestions. By responding to your comments and the comments of the other referees, we believe our manuscript has significantly improved. You will find your original comments in bold, our response to your comments immediately afterwards, and then the sections of revised text, as appropriate. 

The authors did a good job revising their research report. I feel that most of my concerns from the last round of reviews were addressed in this revision. 

Thank you! We are glad to hear that we addressed most of your concerns. 

Still, there are some minor issues that I believe should be revisited once more. The places to be revisited are the following:

 On page 8, lines 176-177, and on page 15, line 281: Next to N, effect size, and power, the alpha-error should be explicitly stated here as well. This is necessary because the authors apply multiple alpha-error criteria throughout their text (see, e.g., the table notes of Tables 1 and 2).

Thank you for this comment. We added the alpha-error in both places in the manuscript. 

[Page 8]: “To conduct the simulation, we assumed a standardized effect size of d=0.30 with 80% power (i.e., β=0.80 and α=0.05).”

[Page 15]: “We conducted an ex-post power analysis to determine the size of the testimonial effect that we could detect with 80% power (with alpha of 0.05) using our sample of 308 participants.”

 On page 14: Why are the expressions “Testimonial” and “Price and Testimonial” capitalized in line 259?

These two expressions represent variable names and appear in text exactly the same as they do in Table 2. In other words, we keep the capitalization so that it is clear that we are referring to the variables’ names from Table 2.

 In response to Reviewer #2’s criticism that the title was too general (“Encouraging pro-environmental behavior”), I suggest the authors seize the opportunity on page 14, lines 265-266, to strengthen their claim. Instead of “This result is intuitive since planting native species represents pro-environmental behavior,” I suggest they write something along the lines of “This result supports our notion that planting native species represents not only a way to express people’s appreciation of the (local) natural environment but also a way to protect the environment (i.e., a pro-environmental behavior).” More details on the argument can be found in Kaiser et al. (2013).

We appreciate this suggestion to strengthen our claim. We included the following sentence on page 14: 

“This result supports our notion that planting native species represents not only a way to express one’s appreciation of the (local) natural environment but also a way to protect the environment (i.e., represents a pro-environmental behavior).”

On page 15, line 289: “skewed to the left” is less informative than “skewed toward not paying anything at all.”

We edited this sentence as you suggested. 

[Page 15]: “The results of these two studies suggest that the mean and standard deviation of values are similar in magnitude and that the distribution of values is skewed toward not paying anything at all.”

I suggest moving the text from page 14 (line 278) to page 17 (line 329) in the Discussion section: before line 353 (now on page 18).

We appreciate this suggestion. We moved the discussion about the potential reasons that could explain why the expert testimonial treatment had no effect to the Discussion section of the manuscript. However, we believe it is important to keep the power analysis discussion (lines 274-301) immediately after the text that explains our findings in the Results section so that the findings are interpreted in the context of the power analysis. The Discussion section now reads as follows: 

[Page 17]: “In this particular context, expert testimonials were not effective at motivating behavioral change, but this research design does not enable us to know why testimonials were not effective. Recent papers have emphasized the importance of identifying the key determinants limiting a desired behavior and then identifying the behavioral interventions that will likely be most effective at targeting those determinants [13] or the mechanisms of action through which changes in behavior occur [64]. One potential explanation as to why the testimonial failed is that it did not effectively target the key barriers limiting adoption of native plants. Previous studies have identified numerous barriers that limit planting of native plants in residential landscapes. These barriers include limited knowledge and information [65], lack of availability [66,67], preferences for particular aesthetics [68], social/community norms [69,65], direct costs and the time and effort required for maintenance and care [67], and characteristics of residents and their yards [70,71]. Expert testimonials may be effective at addressing some of these barriers but not others. 

Following the mapping of determinants and interventions by van Valkengoed et al. [13], we suggest that expert testimonials are generally designed to promote knowledge, change attitudes toward a pro-environmental behavior, and influence injunctive norms by having a respected expert communicating approval of an action. If knowledge, attitudes about integrating native plants in landscaping, and injunctive norms are the main barriers to adoption of native plants, our results indicate that testimonials are not effective at overcoming these barriers in this context. However, it is also possible that our testimonial intervention was not well-matched to the behavioral determinant limiting adoption of native plants. 

Testimonials are not designed to overcome all behavioral barriers. For example, they are likely a poor match to address barriers related to the time and effort required to plant and maintain the native plants or to overcome a general dearth of pro-environmental attitudes. In our experiment, we found that 20% of people did not want to take the native plant even when it given away for free. One factor driving this outcome may be a perception of transaction costs associated with planting (e.g., finding or clearing the area where to plant, time and effort required to plant and care for the plants, etc.) among some participants. Transaction costs have been shown to be a barrier to adoption of pro-environmental behavior in other contexts, such as adoption of agricultural conservation practices [57,58] and residential landscape best management practices [59], among others. Further, without information about the level of the participants’ commitment to environmental protection, it is unclear whether the transaction costs or insufficient strength of environmental attitudes to offset these costs [60,61]. These behavioral drivers are unlikely to be influenced by an expert testimonial treatment; therefore, our results may reflect a general mismatch between the behavioral intervention and the primary barriers limiting adoption of native plants. 

Finally, while we used randomized assignment to the treatment and control groups, for the characteristics that were not balanced across these groups, other unidentified confounding factors (e.g., group differences in environmental attitudes) may have potentially precluded the identification of a statistically significant treatment effect. Further, the participants in our experiment tended to be more educated compared to the general population in the local area and could have already possessed some of the knowledge communicated in the testimonial, which could have led to the lack of a significant impact of the testimonial on participants’ preferences.”

References: 

13. van Valkengoed AM, Abrahamse W, Steg L. To select effective interventions for pro-environmental behaviour change, we need to consider determinants of behaviour. Nature Human Behaviour. 2022 Nov 16:1-1. Available from: https://doi.org/10.1038/s41562-022-01473-w

57. McCann L, Claassen R. Farmer transaction costs of participating in federal conservation programs: Magnitudes and determinants. Land Economics. 2016 May 1;92(2):256-72. Available from: https://doi.org/10.3368/le.92.2.256

58. Palm‐Forster LH, Swinton SM, Lupi F, Shupp RS. Too burdensome to bid: Transaction costs and pay‐for‐performance conservation. American Journal of Agricultural Economics. 2016 Oct;98(5):1314-33. Available from: https://doi.org/10.1093/ajae/aaw071

59. Johnston RJ, Ndebele T, Newburn DA. Modeling transaction costs in household adoption of landscape conservation practices. American Journal of Agricultural Economics. 2023 Jan;105(1):341-67. Available from: https://doi.org/10.1111/ajae.12319

60. Kaiser FG, Kibbe A, Hentschke L. Offsetting behavioral costs with personal attitudes: A slightly more complex view of the attitude-behavior relation. Personality and Individual Differences. 2021 Dec 1;183:111158. Available from: https://doi.org/10.1016/j.paid.2021.111158

61. Kaiser FG, Byrka K, Hartig T. Reviving Campbell’s paradigm for attitude research. Personality and Social Psychology Review. 2010 Nov;14(4):351-67. Available from: https://doi.org/10.1177/1088868310366452

64. Michie S, Carey RN, Johnston M, Rothman AJ, De Bruin M, Kelly MP, Connell LE. From theory-inspired to theory-based interventions: a protocol for developing and testing a methodology for linking behaviour change techniques to theoretical mechanisms of action. Annals of behavioral medicine. 2018 Jun;52(6):501-12. Available from: https://doi.org/10.1007/s12160-016-9816-6

65. Uren HV, Dzidic PL, Bishop BJ. Exploring social and cultural norms to promote ecologically sensitive residential garden design. Landscape and Urban Planning. 2015 May 1;137:76-84. Available from: https://doi.org/10.1016/j.landurbplan.2014.12.008. 

66. Cavender‐Bares J, Padullés Cubino J, Pearse WD, Hobbie SE, Lange AJ, Knapp S, Nelson KC. Horticultural availability and homeowner preferences drive plant diversity and composition in urban yards. Ecological Applications. 2020 Jun;30(4):e02082. Available from: https://doi.org/10.1002/eap.2082.

67. Wheeler MM, Larson KL, Bergman D, Hall SJ. Environmental attitudes predict native plant abundance in residential yards. Landscape and Urban Planning. 2022 Aug 1;224:104443. Available from: https://doi.org/10.1016/j.landurbplan.2022.104443

68. Kendal D, Williams KJ, Williams NS. Plant traits link people's plant preferences to the composition of their gardens. Landscape and urban planning. 2012 Mar 30;105(1-2):34-42. Available from: https://doi.org/10.1016/j.landurbplan.2011.11.023.

69. Goddard MA, Dougill AJ, Benton TG. Why garden for wildlife? Social and ecological drivers, motivations and barriers for biodiversity management in residential landscapes. Ecological economics. 2013 Feb 1;86:258-73. Available from: https://doi.org/10.1016/j.ecolecon.2012.07.016

70. Campbell B, Khachatryan H, Rihn A. Pollinator-friendly plants: Reasons for and barriers to purchase. HortTechnology. 2017 Dec 1;27(6):831-9. Available from: https://doi.org/10.21273/HORTTECH03829-17

71. Larson KL, Lerman SB, Nelson KC, Narango DL, Wheeler MM, Groffman PM, Hall SJ, Grove JM. Examining the potential to expand wildlife-supporting residential yards and gardens. Landscape and Urban Planning. 2022 Jun 1;222:104396. Available from: https://doi.org/10.1016/j.landurbplan.2022.104396

The DOI addresses are missing from several of the articles listed in the Reference list.

Conclusion. 

Thank you for pointing this out. We added DOI where it was missing in the reference list. 

After revisiting these issues and amending the text along the suggested lines, I believe the research report will make a fine contribution to the literature. I hope the article will receive the attention it deserves.

 We appreciate your valuable comments that have improved this manuscript! 

REFERENCES

Kaiser, F. G., Hartig, T., Brügger, A., & Duvier, C. (2013). Environmental protection and nature as distinct attitudinal objects: An application of the Campbell paradigm. Environment and Behavior, 45, 369-398.

 

Author’s response to comments by Reviewer #3: PONE-D-22-27253R1

Title: "Encouraging pro-environmental behavior: Do testimonials by experts work"

Thank you for reading our manuscript and for the helpful comments and suggestions. By responding to your comments and the comments of the other referees, we believe our manuscript has significantly improved. You will find your original comments in bold, our response to your comments immediately afterwards, and then the sections of revised text, as appropriate. 

In this article, the authors conduct an experiment to examine whether an expert testimonial can motivate people to purchase native plants, a behaviour that is more pro-environmental than purchasing non-native plants. The authors found that there was no statistically significant difference in willingness to pay between the people who listened to the expert testimonial and the people in the control condition. The authors conclude that testimony-based interventions alone may not necessarily be sufficient to motivate changes in behaviour.

This paper reports an interesting field experiment with ecological validity. I think the focus on testimonials adds something new to the literature, and I appreciate the measure of behaviour and the relatively realistic research setting (compared to the traditional lab experiment). I am looking at this work for the first time now, but I realise the authors have already addressed many comments from the previous round of reviews, and seem to have done so thoroughly.

My addition to the discussion about this paper does not pertain to the methodology or the results, but more to how this research fits in with overall discussion on behaviour change interventions for pro-environmental behaviour. In recent years, we have seen many meta-analyses and review articles that tried to find out which interventions to promote pro-environmental behaviour are effective and which are not (e.g., Byerly et al., 2018; Mertens et al., 2022; Nisa et al., 2019). This approach of examining which interventions are effective overall has been criticized, since it is unlikely that particular interventions will always be effective whereas others will always be ineffective. Rather, the effectiveness of an intervention depends on the extent to which the intervention matches the target behaviour and the context in which the behaviour is implemented (van Valkengoed et al., 2022). This is broadly recognized within the field of behaviour change in general as well (e.g., Michie et al., 2018).

In general, I find that this research is not well embedded in this ongoing discussion in the literature, and does not really engage with any of the previously mentioned articles. In addition, this research seems to fall in line with the philosophy of trying to determine whether an intervention is effective or not, without considering the context in which the intervention is implemented, and whether there is a match between the intervention and the target behaviour. For example, the authors currently draw the conclusion that ‘in pro-environmental contexts, testimonial-based interventions alone may not necessarily be sufficient to motivate changes in behavior’. I would like to ask the authors to move away from these general conclusions about whether interventions do or do not work in general. Rather, I would like the authors to reflect more on why a testimonial-based intervention did not work in this specific context.

Following the procedure outlined by van Valkengoed et al., (2022), this would involve a more extensive discussion on what the possible drivers and barriers are to purchasing non-native plants, and an analysis of which of these barriers are and are not addressed by the intervention in question. Throughout the article, the authors already highlight several relevant barriers to purchasing native plants, such as landscaping preferences and being unwilling to assume the cost and/or care of the plant. Given the apparent presence of these barriers, it is not surprising to me that the testimonial intervention did not work in this context, since an expert testimonial does not address these specific barriers directly. There seems to be a mismatch between the intervention and the barriers to the target behaviour. I think this would be a fairer interpretation of the findings in this paper compared to drawing conclusions about testimonial-based interventions in general, especially since the authors themselves cite many papers in the introduction of this paper that highlight the effectiveness of testimony-interventions. As the paper currently stands, this leaves the reader wondering why the authors conclude that testimonials do not work for pro-environmental behaviours.

References

Byerly, H., Balmford, A., Ferraro, P. J., Hammond Wagner, C., Palchak, E., Polasky, S., . . Fisher, B. (2018). Nudging pro-environmental behavior: evidence and opportunities. Frontiers in Ecology and the Environment, 16(3), 159-168. https://doi.org/10.1002/fee.1777

Mertens, S., Herberz, M., Hahnel, U. J. J., & Brosch, T. (2022). The effectiveness of nudging: A meta-analysis of choice architecture interventions across behavioral domains. Proc Natl Acad Sci U S A, 119(1). https://doi.org/10.1073/pnas.2107346118

Michie, S., Carey, R. N., Johnston, M., Rothman, A. J., de Bruin, M., Kelly, M. P., & Connell, L. E. (2018). From Theory-Inspired to Theory-Based Interventions: A Protocol for Developing and Testing a Methodology for Linking Behaviour Change Techniques to Theoretical Mechanisms of Action. Annals of Behavioral Medicine, 52(6), 501-512. https://doi.org/10.1007/s12160-016-9816-6

Nisa, C. F., Belanger, J. J., Schumpe, B. M., & Faller, D. G. (2019). Meta-analysis of randomised controlled trials testing behavioural interventions to promote household action on climate change. Nat Commun, 10(1), 4545. https://doi.org/10.1038/s41467-019-12457-2

van Valkengoed, A. M., Abrahamse, W., & Steg, L. (2022). To select effective interventions for pro-environmental behaviour change, we need to consider determinants of behaviour. Nat Hum Behav. https://doi.org/10.1038/s41562-022-01473-w

We appreciate your comments and insights. In the discussion section of the revised manuscript, we have sought to better align the insights of this research to the ongoing discussion in the papers that you have suggested. As you note, this paper has already gone through multiple revisions with other reviewers and the editor who view this paper as a positive contribution to the literature. Thus, we decided not to overhaul the framing of the paper as the other reviewers have endorsed the current framing. Hopefully our approach of using the discussion to address the important issues you raise in your comments strikes a good balance. 

[Page 17]: “In this particular context, expert testimonials were not effective at motivating behavioral change, but this research design does not enable us to know why testimonials were not effective. Recent papers have emphasized the importance of identifying the key determinants limiting a desired behavior and then identifying the behavioral interventions that will likely be most effective at targeting those determinants [13] or the mechanisms of action through which changes in behavior occur [64]. One potential explanation as to why the testimonial failed is that it did not effectively target the key barriers limiting adoption of native plants. Previous studies have identified numerous barriers that limit planting of native plants in residential landscapes. These barriers include limited knowledge and information [65], lack of availability [66,67], preferences for particular aesthetics [68], social/community norms [69,65], direct costs and the time and effort required for maintenance and care [67], and characteristics of residents and their yards [70,71]. Expert testimonials may be effective at addressing some of these barriers but not others. 

Following the mapping of determinants and interventions by van Valkengoed et al. [13], we suggest that expert testimonials are generally designed to promote knowledge, change attitudes toward a pro-environmental behavior, and influence injunctive norms by having a respected expert communicating approval of an action. If knowledge, attitudes about integrating native plants in landscaping, and injunctive norms are the main barriers to adoption of native plants, our results indicate that testimonials are not effective at overcoming these barriers in this context. However, it is also possible that our testimonial intervention was not well-matched to the behavioral determinant limiting adoption of native plants. 

Testimonials are not designed to overcome all behavioral barriers. For example, they are likely a poor match to address barriers related to the time and effort required to plant and maintain the native plants or to overcome a general dearth of pro-environmental attitudes. In our experiment, we found that 20% of people did not want to take the native plant even when it given away for free. One factor driving this outcome may be a perception of transaction costs associated with planting (e.g., finding or clearing the area where to plant, time and effort required to plant and care for the plants, etc.) among some participants. Transaction costs have been shown to be a barrier to adoption of pro-environmental behavior in other contexts, such as adoption of agricultural conservation practices [57,58] and residential landscape best management practices [59], among others. Further, without information about the level of the participants’ commitment to environmental protection, it is unclear whether the transaction costs or insufficient strength of environmental attitudes to offset these costs [60,61]. These behavioral drivers are unlikely to be influenced by an expert testimonial treatment; therefore, our results may reflect a general mismatch between the behavioral intervention and the primary barriers limiting adoption of native plants.” 

[Page 19]: “Future studies can extend this research by identifying the primary barriers limiting adoption of native plants and testing behavioral interventions that are best suited to overcome these determinants [13,64]. To further inform policymakers and program administrators, future studies can also test the efficacy of testimonial-based interventions in the context of other pro-environmental behavior in which the barriers are related to knowledge, attitudes, and injunctive norms – behavioral determinants that are more likely to be addressed by a testimonial intervention. Additionally, examining the effect of combining testimonials with other non-monetary behavioral interventions and/or financial incentives, or using larger samples to detect smaller effects of testimonials alone may be fruitful directions for future research.”

References: 

13. van Valkengoed AM, Abrahamse W, Steg L. To select effective interventions for pro-environmental behaviour change, we need to consider determinants of behaviour. Nature Human Behaviour. 2022 Nov 16:1-1. Available from: https://doi.org/10.1038/s41562-022-01473-w

57. McCann L, Claassen R. Farmer transaction costs of participating in federal conservation programs: Magnitudes and determinants. Land Economics. 2016 May 1;92(2):256-72. Available from: https://doi.org/10.3368/le.92.2.256

58. Palm‐Forster LH, Swinton SM, Lupi F, Shupp RS. Too burdensome to bid: Transaction costs and pay‐for‐performance conservation. American Journal of Agricultural Economics. 2016 Oct;98(5):1314-33. Available from: https://doi.org/10.1093/ajae/aaw071

59. Johnston RJ, Ndebele T, Newburn DA. Modeling transaction costs in household adoption of landscape conservation practices. American Journal of Agricultural Economics. 2023 Jan;105(1):341-67. Available from: https://doi.org/10.1111/ajae.12319

60. Kaiser FG, Kibbe A, Hentschke L. Offsetting behavioral costs with personal attitudes: A slightly more complex view of the attitude-behavior relation. Personality and Individual Differences. 2021 Dec 1;183:111158. Available from: https://doi.org/10.1016/j.paid.2021.111158

61. Kaiser FG, Byrka K, Hartig T. Reviving Campbell’s paradigm for attitude research. Personality and Social Psychology Review. 2010 Nov;14(4):351-67. Available from: https://doi.org/10.1177/1088868310366452

64. Michie S, Carey RN, Johnston M, Rothman AJ, De Bruin M, Kelly MP, Connell LE. From theory-inspired to theory-based interventions: a protocol for developing and testing a methodology for linking behaviour change techniques to theoretical mechanisms of action. Annals of behavioral medicine. 2018 Jun;52(6):501-12. Available from: https://doi.org/10.1007/s12160-016-9816-6

65. Uren HV, Dzidic PL, Bishop BJ. Exploring social and cultural norms to promote ecologically sensitive residential garden design. Landscape and Urban Planning. 2015 May 1;137:76-84. Available from: https://doi.org/10.1016/j.landurbplan.2014.12.008. 

66. Cavender‐Bares J, Padullés Cubino J, Pearse WD, Hobbie SE, Lange AJ, Knapp S, Nelson KC. Horticultural availability and homeowner preferences drive plant diversity and composition in urban yards. Ecological Applications. 2020 Jun;30(4):e02082. Available from: https://doi.org/10.1002/eap.2082.

67. Wheeler MM, Larson KL, Bergman D, Hall SJ. Environmental attitudes predict native plant abundance in residential yards. Landscape and Urban Planning. 2022 Aug 1;224:104443. Available from: https://doi.org/10.1016/j.landurbplan.2022.104443

68. Kendal D, Williams KJ, Williams NS. Plant traits link people's plant preferences to the composition of their gardens. Landscape and urban planning. 2012 Mar 30;105(1-2):34-42. Available from: https://doi.org/10.1016/j.landurbplan.2011.11.023.

69. Goddard MA, Dougill AJ, Benton TG. Why garden for wildlife? Social and ecological drivers, motivations and barriers for biodiversity management in residential landscapes. Ecological economics. 2013 Feb 1;86:258-73. Available from: https://doi.org/10.1016/j.ecolecon.2012.07.016

70. Campbell B, Khachatryan H, Rihn A. Pollinator-friendly plants: Reasons for and barriers to purchase. HortTechnology. 2017 Dec 1;27(6):831-9. Available from: https://doi.org/10.21273/HORTTECH03829-17

71. Larson KL, Lerman SB, Nelson KC, Narango DL, Wheeler MM, Groffman PM, Hall SJ, Grove JM. Examining the potential to expand wildlife-supporting residential yards and gardens. Landscape and Urban Planning. 2022 Jun 1;222:104396. Available from: https://doi.org/10.1016/j.landurbplan.2022.104396

---

## [Decision Letter · Decision Letter 2]

27 Jul 2023

PONE-D-22-27253R2Encouraging pro-environmental behavior: do testimonials by experts work?PLOS ONE

Dear Dr. Savchenko,

Thank you for submitting your manuscript to PLOS ONE. After careful consideration, we feel that it has merit but does not fully meet PLOS ONE’s publication criteria as it currently stands. Therefore, we invite you to submit a revised version of the manuscript that addresses the points raised during the review process. I invite you to incorporate the minor points pointed out by reviewer 1. After this consideration of the smaller points, I assume that nothing should stand in the way of the publication.

We look forward to receiving your revised manuscript.

Kind regards,

Prof. Dr. Florian Follert

Academic Editor

PLOS ONE

Journal Requirements:

Reviewers' comments:

Reviewer's Responses to Questions

**Comments to the Author**

1. If the authors have adequately addressed your comments raised in a previous round of review and you feel that this manuscript is now acceptable for publication, you may indicate that here to bypass the “Comments to the Author” section, enter your conflict of interest statement in the “Confidential to Editor” section, and submit your "Accept" recommendation.

Reviewer #1: (No Response)

Reviewer #3: All comments have been addressed

2. Is the manuscript technically sound, and do the data support the conclusions?

Reviewer #1: Yes

Reviewer #3: Yes

3. Has the statistical analysis been performed appropriately and rigorously? 

Reviewer #1: Yes

Reviewer #3: Yes

4. Have the authors made all data underlying the findings in their manuscript fully available?

Reviewer #1: (No Response)

Reviewer #3: Yes

5. Is the manuscript presented in an intelligible fashion and written in standard English?

Reviewer #1: Yes

Reviewer #3: (No Response)

6. Review Comments to the Author

Reviewer #1: REVIEW OF PONE-D-22-27253.R2

"Encouraging pro-environmental behavior: Do testimonials by experts work?"

Once again, the authors did a good job revising their research report. Overall, they followed my suggestions rather closely (with only one exception). They stated that they changed “skewed to the left” to “skewed toward not paying anything at all” (p. 15, line 285), but this change was not actually made.

On second thought, I now feel that Figure 4 is rather uninformative and that it would be better to replace it with a sentence that reports the two numbers plus confidence intervals.

Moreover, I suggest that on p. 18 the authors replace the terms “pro-environmental attitudes” (line 346) and “environmental attitudes” (lines 355 & 361) with “commitment to protecting the environment.” The term attitude can be misleading, as it sometimes refers to verbally expressed opinions and sometimes to people’s propensity or, as I prefer, to people’s commitment to act in a particular way (e.g., to protect the environment).

The references are typically numbered in order of their appearance, but this convention is sometimes violated (see pp. 17 & 18).

Conclusion. If the authors revisit the remaining small issues and amend the text along the lines I have suggested, the research report will make a fine contribution to the literature. Again, I hope the article will receive the attention it deserves.

Reviewer #3: I think the new text that the authors have added to the discussion perfectly captures the point I tried to make in my review last time. I appreciate how the authors now discuss this issue of matching the intervention to relevant determinants in detail, and also cite previous studies that give an overview of the relevant determinants that play a role in sustainable gardening choices. I commend the authors on this effort and have no further comments on this manuscript.

7. PLOS authors have the option to publish the peer review history of their article (what does this mean?). If published, this will include your full peer review and any attached files.

Reviewer #1: **Yes: **Florian G Kaiser

Reviewer #3: No

---

## [Author Response · Author response to Decision Letter 2]

30 Aug 2023

Once again, the authors did a good job revising their research report. Overall, they followed my suggestions rather closely (with only one exception). They stated that they changed “skewed to the left” to “skewed toward not paying anything at all” (p. 15, line 285), but this change was not actually made.

Thank you for catching this! We updated this language in the manuscript accordingly. 

On second thought, I now feel that Figure 4 is rather uninformative and that it would be better to replace it with a sentence that reports the two numbers plus confidence intervals.

We agree. We took out Figure 4 and replaced it with the language below. 

[p. 14]: “Using equation 2 and the coefficient estimates from Model 2, we calculated the mean WTP for native plants for the control and treatment groups. Among the participants in the treatment group, the mean WTP for native plants was $3.28, 95% CI [2.66, 3.90], showing a $0.26 increase compared to the mean WTP of $3.02, 95% CI [2.39, 3.66] among the participants in the control group. This difference, however, is not statistically significant, which is consistent with the results of our earlier analysis.”

Moreover, I suggest that on p. 18 the authors replace the terms “pro-environmental attitudes” (line 346) and “environmental attitudes” (lines 355 & 361) with “commitment to protecting the environment.” The term attitude can be misleading, as it sometimes refers to verbally expressed opinions and sometimes to people’s propensity or, as I prefer, to people’s commitment to act in a particular way (e.g., to protect the environment).

[p. 18]: 

“For example, they are likely a poor match to address barriers related to the time and effort required to plant and maintain the native plants or to overcome a general dearth of efforts to protect the environment.”

“Further, without information about the level of the participants’ commitment to environmental protection, it is unclear whether the transaction costs or lack of sufficient commitment to environmental protection to offset these costs [69,70] may have led to the ineffectiveness of the testimonial treatment.”

“Finally, while we used randomized assignment to the treatment and control groups, for the characteristics that were not balanced across these groups, other unidentified confounding factors (e.g., group differences in commitment to protect the environment) may have potentially precluded the identification of a statistically significant treatment effect.”

The references are typically numbered in order of their appearance, but this convention is sometimes violated (see pp. 17 & 18).

Thank you for pointing this out. We updated the reference numbers accordingly. 

Conclusion. If the authors revisit the remaining small issues and amend the text along the lines I have suggested, the research report will make a fine contribution to the literature. Again, I hope the article will receive the attention it deserves.

Thank you for your careful review of our work. We appreciate the time you have invested in providing us with valuable feedback, which has improved this manuscript.

---

## [Editor Report · Decision Letter 3]

4 Sep 2023

Encouraging pro-environmental behavior: Do testimonials by experts work?

PONE-D-22-27253R3

Dear Dr. Savchenko,

We’re pleased to inform you that your manuscript has been judged scientifically suitable for publication and will be formally accepted for publication once it meets all outstanding technical requirements.

Kind regards,

Florian Follert

Academic Editor

PLOS ONE
---

## [Editor Report · Acceptance letter]

12 Sep 2023

PONE-D-22-27253R3 

Encouraging pro-environmental behavior:
Do testimonials by experts work? 

Dear Dr. Savchenko:

I'm pleased to inform you that your manuscript has been deemed suitable for publication in PLOS ONE. Congratulations! Your manuscript is now with our production department. 

Kind regards, 

on behalf of

Prof. Dr. Florian Follert 

Academic Editor

PLOS ONE